# Maturase K forms a plastidial splicing complex with a neofunctionalized branching enzyme

Yuanyuan Liang [1], Yang Gao [2], Andrea Fontana[1], Melanie Abt[1], Adam Gicgier [1,3], Muriel Gehring [1], Chun Liu[1,4], Mayank Sharma [1], Reimo Zoschke [2], Samuel C. Zeeman [1] & Barbara Pfister [1] ✉

Chloroplast group IIA introns originate from bacterial ribozymes. Their splicing requires the splicing factor Maturase K (MatK). MatK, however, has been difficult to functionally analyze, as it appears essential for plant viability and is encoded in the chloroplast genome. Here we identified a heteromultimeric complex comprising MatK and three other essential, plastid-targeted proteins using co-immunoprecipitation experiments in Arabidopsis and tobacco. Among the MatK interactors is a conserved homologue of starch-branching enzymes (BEs), which we named MATURASE K INTERACTING PROTEIN1 (MKIP1). We demonstrate that MKIP1 proteins have lost BE activity and acquired a 150-amino acid insertion that enables direct interaction with MatK's N-terminus. Immunoprecipitation of Arabidopsis MKIP1 co-precipitates all known MatK intron targets. Inducing *MKIP1* silencing in Arabidopsis causes newly emerging leaves to be pale, in which the splicing of MatK intron targets is strongly reduced. Our data suggest that MKIP1 functionally diverged from canonical BEs to facilitate splicing in conjunction with MatK. In turn, the former reverse-transcriptase domain in the N-terminal region of MatK likely has acquired the capacity to interact with other proteins. Potentially, complex formation allowed MatK to diversify its RNA interactions, helping its transition towards a general splicing factor.

Chloroplast genomes (plastomes) of land plants contain ~20 introns. As they interrupt the sequences of genes required for photosynthesis, protein homeostasis, and for plastidial gene expression itself, intron splicing is an indispensable stage of plastidial RNA processing and its failure usually results in plant inviability[1,2]. Most plastidial introns belong to group II introns derived from mobile bacterial elements, which are also probable predecessors of nuclear mRNA introns[3]. Much of our understanding of the mechanism of plastidial intron splicing derives from analogy to the splicing of these bacterial ancestors. All group II introns share an RNA structure consisting of six double-helical

domains (denoted DI-DVI)[4]. In bacteria, these were shown to fold into a complex tertiary structure that allows the RNA to splice via two consecutive splicing steps, resulting in a branched intron lariat and ligated exons[5]. However, although splicing is catalyzed by the ribozyme intron RNA itself, efficient splicing of bacterial introns usually requires a maturase protein[6]. These are typically encoded within intron domain IV and have an N-terminal reverse transcriptase (RT) domain, followed by a maturase (X) domain, among others[7]. During splicing, the RT domain of the bacterial maturase stably docks the protein onto the intron in a sequence-specific manner, allowing the X domain to make

[1]Institute of Molecular Plant Biology, ETH Zurich, Auguste-Piccard-Hof 1, Zurich, Switzerland. [2]Max Planck Institute of Molecular Plant Physiology, Am Mühlenberg 1, Potsdam-Golm, Germany. [3]Present address: Department of Crop Genetics, John Innes Centre, Norwich, UK. [4]Present address: Crop Science Centre, University of Cambridge, Lawrence Weaver Rd, Cambridge, UK. ✉e-mail: barbara.pfister@biol.ethz.ch

flexible contacts with the intron core[7–9]. The X domain then facilitates critical RNA conformations during splicing by stabilizing the intron's catalytic core and the position of the bulged adenosine in domain VI, the nucleophile of the first splicing step[10,11].

Bacterial group II introns each encode their own, highly specific maturase. Contrarily, in land plants chloroplasts, only the *trnK* intron encodes a maturase-like protein, Maturase K (MatK), while the other group II introns do not. Tobacco MatK was observed to bind to six or seven introns, i.e., to all group IIA introns except the divergent introns in *clpP* and (possibly) *trnV*[12–15]. MatK likely helps the splicing of its binding targets, since splicing is reduced upon MatK reduction by genetic or chemically-induced disruption of plastid translation, and since *matK* gene loss coincides with intron loss in several parasitic plants[16]. These observations imply that MatK has evolved into a general splicing factor, but direct functional analyses of MatK have been hindered by the lack of (hypomorphic) mutants of this likely essential gene[12].

Consistent with a potential functional diversification, MatK has structurally diverged from bacterial maturases. It retained its X domain, indicating that it can still interact with the intron core facilitating intron splicing[17]. However, its RT domain appears degenerated, since the N-terminal RT motifs of bacterial maturases are no longer recognizable[17]. This includes most of RT0 region which enables bacterial maturases to form strong and sequence-specific interactions with intron domain IV[8,9,18]. The potential loss of specific RNA binding may have helped MatK to act on multiple introns but likely comes at the cost of a lower RNA affinity and may negatively affect its orientation on the intron[7]. In addition, the plastidial introns have lost RNA-RNA interaction motifs involved in the folding of bacterial introns[19]. These deviations may explain why plastidial group IIA intron splicing requires an array of nuclear-encoded splicing factors, such as CRS1 (Chloroplast RNA Splicing 1), RNC1 and WTF1 ("what's this factor?")[20–22]. These possibly assist in the initial folding and stabilization of the precursor[1,2], but have not been reported to associate directly with MatK.

Here we report a protein that physically interacts with MatK to help plastidial intron splicing in *Arabidopsis thaliana* (Arabidopsis). This protein was previously named BRANCHING ENZYME1 (*At*BE1) (AT3G20440) based on its homology to starch-branching enzymes[23] (BEs), which introduce α−1,6-linkages into the polymers comprising starch, a plastidial storage carbohydrate[24]. However, in contrast to the other two Arabidopsis BEs (*At*BE2 and *At*BE3), *At*BE1 has no apparent BE activity[23,25]. In addition, Arabidopsis *be1* mutants are embryo defective and inviable[26,27], while *be2* and *be3* single mutants are phenotypically normal[23]. The *be2/be3* double mutant lacks starch and grows poorly but is viable[23], as are, to our knowledge, all other Arabidopsis mutants with primary defects in starch metabolism described to date. We show that *At*BE1 is a conserved, neofunctionalized BE that has acquired a new role in plastid intron splicing by forming a splicing complex with *At*MatK. To recognize its function and avoid confusion with canonical BEs, we renamed it to MATURASE K INTERACTING PROTEIN1, or MKIP1.

## Results

### *At*MKIP1 is required for plant viability

We reconfirmed that *At*MKIP1 is essential for embryo development[26,27] using two lines (*mkip1-1* and *mkip1-2*) carrying T-DNA insertions in introns 15 and 16 of the *AtMKIP1* (AT3G20440) gene, respectively (Fig. 1a, Supplementary Fig. 1). No *mkip1*[-/-] mutant plants could be identified from the progeny of mother plants heterozygous for either allele (192 and 100 plants analyzed for *mkip1-1* and *mkip1-2*, respectively). In addition, siliques from heterozygous plants contained ~25% white seeds with embryos arrested at the heart stage (Fig. 1a, Supplementary Fig. 1). These phenotypes could be complemented by $P_{UBQ10}$-driven, constitutive expression of the coding sequence (CDS) of

*At*MKIP1 fused to a C-terminal mCitrine or eYFP (named YFP hereafter) tag in heterozygous *mkip1-1* or *mkip1-2* plants. Complemented *mkip1-1*[-/-] and *mkip1-2*[-/-] plants appeared wild-type (WT) like, although they grew slightly slower (Fig. 1, Supplementary Figs. 1, 2c). Confocal microscopy of $P_{35S}$-driven, stable expression of *At*MKIP1-YFP in WT Arabidopsis confirmed that *At*MKIP1 exclusively localizes to the chloroplasts of epidermal cells and shows a fairly homogeneous distribution within the chloroplast stroma[26] (Fig. 1b, Supplementary Fig. 3).

In an attempt to bypass the embryonic arrest of *mkip1*[-/-] mutants, we transformed heterozygous *mkip1-1*[+/-] and *mkip1-2*[+/-] plants with a construct expressing *At*MKIP1-mCitrine under the control of the embryo-specific *ABI3* promoter[28]. Siliques from *mkip1*[+/-] plants homozygous for the construct did not contain any white seeds, indicating a successful rescue of the embryo-defectiveness of *mkip1*[-/-] (Fig. 1a, Supplementary Fig. 1d). We recovered rescued seedlings in both *mkip1-1*[-/-] and *mkip1-2*[-/-] backgrounds at the expected frequency of ~25%. However, after germination, these seedlings developed serrated, white true leaves and abnormal meristems and were seedling lethal (Fig. 1c, Supplementary Fig. 1e). These phenotypes were similar to $P_{ABI3}$-rescue lines of mutants severely impaired in chloroplast differentiation due to primary defects in plastidial translation or splicing[29]. Seedling lethality of rescued *mkip1-1*[-/-] could not be rescued by growing plants on ½-strength MS medium supplemented with 1% sucrose (Supplementary Fig. 1c).

Plastids from true leaves of $P_{ABI3}$-rescued *mkip1-1*[-/-] seedlings mostly contained collapsed thylakoid stacks and accumulated electron-dense, vesicular structures of unknown nature. In greenish leaf areas, we observed chloroplast-like plastids with large condensed thylakoid stacks (Fig. 1e, outer right micrograph), reminiscent of structures observed upon co-suppression of the plastidial ribosomal protein *At*bS1c (RPS1)[30]. The cotyledons were pale green, and their plastids had abnormally stacked thylakoids (Fig. 1c, e). Presumably, the cotyledons were less affected due to residual *At*MKIP1-mCitrine protein remaining in cotyledon tissue from embryogenesis, although *At*MKIP1-mCitrine levels were below the detection limit (Fig. 1d). These phenotypes strongly suggest that *At*MKIP1 is essential for plant viability also after embryogenesis and are consistent with a putative essential role of *At*MKIP1 in chloroplast function, but not in starch metabolism.

### *At*MKIP1 and orthologs have lost branching enzyme activity

Phylogenetic analyses showed that the canonical BEs involved in starch biosynthesis of land plants and streptophyte algae belong to either class I (which is absent in Arabidopsis) or class II (to which *At*BE2 and *At*BE3 belong; Fig. 2a), confirming previous observations[31]. The BEs of chlorophyte algae also mostly belong to these classes, whereas those of glaucophyte and rhodophyte algae, which produce floridean starch in the cytosol[32], form a separate group. *At*MKIP1 and its orthologs form yet another, well-separated clade of class III BEs[31], which share the structural features of *At*MKIP1 described below. We identified at least one class III BE in each photoautotrophic land plant examined and in some streptophyte algae, but not in any other alga, suggesting that it arose during the colonization of land by plants (Fig. 2a, Supplementary Fig. 4a). The origin of class III BEs is unclear, but their distinct exon-intron gene organization[31] renders it unlikely that they arose from the duplication of a class I or II BE gene (Supplementary Fig. 4b).

Class III BEs retain a classical BE domain architecture consisting of a family 48 carbohydrate-binding module (CBM) and α-amylase domains (Fig. 2b). Nonetheless, they display striking differences from canonical BEs, with which they share only ~30% amino acid (AA) identity[31]. First, class III BEs possess an MKIP1-specific region (MSR) of ~150 AA after the CBM. This stretch is enriched for positively and negatively charged AA (especially lysines and glutamates, respectively) and is predicted to form a largely disordered region protruding from

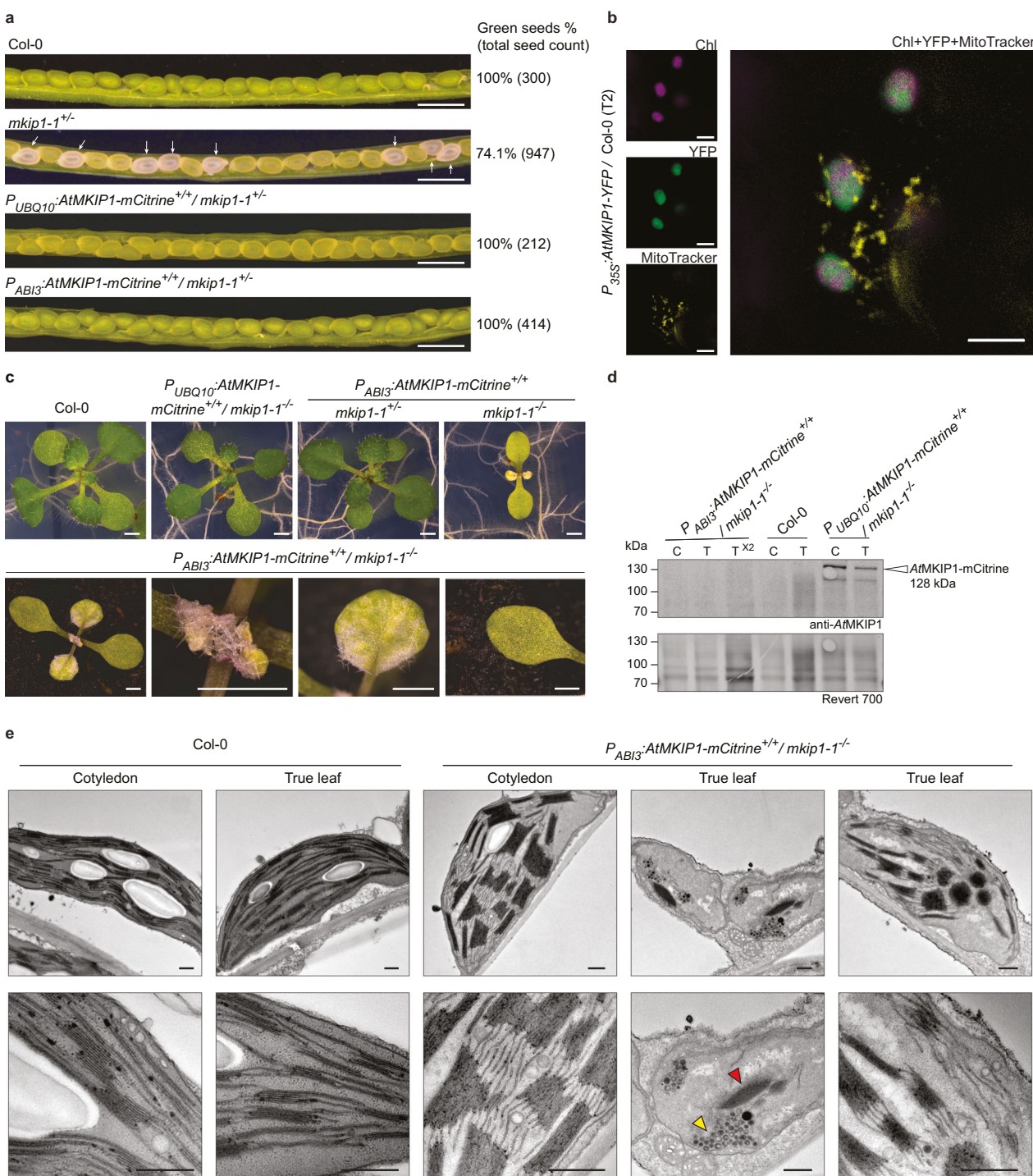

the *At*MKIP1 core (Supplementary Fig. 5). An *At*MKIP1(ΔMSR)-YFP version lacking the MSR failed to complement the embryo defects of *mkip1-1*[-/-], despite being expressed as a soluble protein, indicating a key role for *At*MKIP1 function (Supplementary Fig. 6).

Class III BEs also carry substitutions in several residues required for starch branching catalysis, including two aspartate residues of the catalytic triad[33] (H558 and N682 in *At*MKIP1; Fig. 2b), consistent with *At*MKIP1's apparent absence of BE activity[23,25]. To test whether glucan BE activity is also absent in orthologous proteins, we expressed MKIP1 versions from tobacco (*Nicotiana tabacum*), maize (*Zea mays*), a moss (*Physcomitrium patens*) and a streptophyte alga (*Coleochaete scutata*) as YFP-tagged proteins in the yeast *Saccharomyces cerevisiae* (Fig. 2c). The parental yeast strain used ("578"), contains all components

required for glucan production except a BE, as it has an Arabidopsis gene encoding a starch synthase (*At*SS1) and a bacterial gene for producing its substrate ADPglucose (*Ec*glgC-TM-HA), but lacks its endogenous glycogen BE as well as all other yeast glycogen-metabolic genes[34]. All class III BEs were expressed as soluble proteins in yeast, although in some cases at low levels (Fig. 2d). However, none of them showed enzymatic activity on native BE-activity gels, whereas all tested class I/II BEs did (Fig. 2e). Expression of class III BEs also did not result in glucan synthesis in yeast, as judged by the absence of dark staining after iodine staining of glucans produced in yeast cells (Fig. 2f). The BE activity of *At*MKIP1-YFP could not be restored by reconstituting its former catalytic triad in the *At*MKIP1(H558D,N682D)-YFP version (Fig. 2e, f). These data suggest that the absence of BE activity is a

**Fig. 1 | *At*MKIP1 is essential for embryonic and post-embryonic development.**
**a** Opened siliques of wild type (Col-0), *mkip1-1*[+/-], and *mkip1-1*[+/-] plants transformed with $P_{UBQ10}$:*AtMKIP1-mCitrine* or $P_{ABI3}$:*AtMKIP1-mCitrine* constructs. White seeds are indicated by arrows. Percentages indicate the proportion of green seeds among the total seeds (number of seeds analyzed in parentheses). Scale bars: 1 mm. **b** Confocal light microscopy of leaf epidermal cells of plants stably expressing *At*MKIP1-YFP in wild-type Arabidopsis background. Mitochondria (*yellow*) were stained with Mito-Tracker. Chlorophyll (Chl) autofluorescence is false-colored magenta, and *At*MKIP1-YFP is shown in green. Pixel brightness of individual channels was adjusted uniformly for optimal visualization of signals using ZEN microscopy software. At least 5 images were taken per line, representative micrographs are shown. Chloroplast localization of *At*MKIP1-YFP was also observed in another independent line and in a separate experiment involving three independent lines. Scale bars are 5 μm. **c** *mkip1-1*[-/-] seedlings rescued with the $P_{ABI3}$:*AtMKIP1-mCitrine* construct. Photographs show representative 16-day-old seedlings of the indicated genotypes grown on 1/2-strength MS agar plates. Micrographs of a representative 4-week-old, soil-grown $P_{ABI3}$:*AtMKIP1-mCitrine*[+/+]/ *mkip1-1*[-/-] plant show the whole plant, the

meristem, a true leaf, and a cotyledon (from left to right). Scale bars: 1 mm. Twenty replicate plants and equivalent analyses in the *mkip1-2*[-/-] background yielded similar results (Supplementary Fig. 1). **d** Anti-*At*MKIP1 immunoblot of total proteins from cotyledon (C) or true leaf (T) tissue of the indicated lines. Samples were loaded on an equal fresh weight basis. X2 indicates double loading. The predicted molecular weight of *At*MKIP1-mCitrine refers to the mature proteoform without the chloroplast transit peptide. Native *At*MKIP1 (98 kDa) in Col-0 could not be detected due to insufficient antibody sensitivity. Revert 700 is a total protein stain (loading control). Equivalent analyses in the *mkip1-2*[-/-] background gave similar results. **e** Transmission electron micrographs of chloroplasts from 4-week-old, soil-grown Col-0 and *mkip1-1*[-/-] plants rescued with the $P_{ABI3}$:*AtMKIP1-mCitrine* construct. In the true leaves from rescued *mkip1-1*[-/-] plants, plastids typically contained collapsed thylakoids (red arrow) and vesicular structures (yellow arrow). In greenish areas of the leaf, also larger thylakoid stacks were observed (outer right image). Representative images of at least 7 plastids assessed per line and tissue are shown. Two additional replicate plants and equivalent analyses in the *mkip1-2*[-/-] background gave similar results. Scale bars: 0.5 μm.

general feature of class III BEs and is likely due to widespread sequence divergence.

To test whether the remaining BE-like features in *At*MKIP1 are nevertheless required for its function in Arabidopsis, we mutated a glutamate and a histidine residue essential for canonical BE activity as acid/base catalyst[35] and transition-state stabilizer[36], respectively, which have been retained in MKIP1 proteins (Supplementary Fig. 2a). Expression of this *At*MKIP1(E612A,H681A)-mCitrine construct complemented the *mkip1-1*[-/-] mutant phenotype similarly to the non-mutated *At*MKIP1-mCitrine construct (Supplementary Fig. 2). Complementation was also observed by *At*MKIP1(W151A,W171A)-mCitrine, which carries substitutions in conserved tryptophan residues critical to glucan binding of family-48 CBMs[37,38] (Supplementary Fig. 2). Taken together, these data demonstrate that *At*MKIP1 function predominantly requires others features than canonical BEs, including its newly evolved MSR domain.

## MKIP1 interacts with MatK in an RNA-independent manner
We conducted co-immunoprecipitation (IP) experiments coupled with mass spectrometry (MS) to identify potential interaction partners of *At*MKIP1. *At*MKIP1-YFP stably expressed in Arabidopsis significantly enriched *At*MatK in three independent anti-YFP IPs, each with multiple replicates (Fig. 3a). These experiments further suggested that *At*MKIP1 interacts with two other plastid-localized proteins essential for plant viability: *At*ValRS2 (*At*EMB2247), a putative valyl-tRNA synthetase[39], and *At*EMB3120 (EMBRYO-DEFECTIVE 3120 or SNOW-WHITE LEAF1, SWL1), which is a stromal protein required for plastid development, but of unknown molecular function[40,41]. We confirmed the interactions between *At*MKIP1, *At*ValRS2 and *At*EMB3120 by multiple (reverse-)IPs performed in Arabidopsis (Fig. 3b, Supplementary Fig. 7). In one of these experiments, endogenous *At*EMB3120 was not detectably co-precipitated by *At*MKIP1-YFP. However, overexpressed *At*EMB3120-YFP consistently co-precipitated endogenous *At*MKIP1, suggesting that also *At*EMB3120 is a bona-fide interaction partner of *At*MKIP1 (Supplementary Fig. 7). However, as the functions of *At*ValRS2 and *At*EMB3120 are poorly characterized, we focus here on *At*MatK.

Since *At*MatK is encoded in the plastome, it cannot be routinely manipulated in Arabidopsis, and our attempts to raise a working antibody against *At*MatK were unsuccessful. We thus used *Nt*MatK C+ transplastomic tobacco (*N. tabacum*), which expresses *Nt*MatK with a C-terminal triple HA tag at its native plastome locus[12] (Fig. 3c). The *Nt*MatK-HA protein is functional since it can substitute for *Nt*MatK, which is considered an essential protein[12]. In IP experiments with tobacco seedlings, *Nt*MKIP1 co-precipitated with *Nt*MatK-HA (line *Nt*MatK C + ; Fig. 3d). This interaction was not abolished by treating the IP samples with an RNase A/T1 mixture, which effectively degraded the

RNA (Supplementary Fig. 8), suggesting that the interaction is not bridged by (intron) RNA (Fig. 3d). MS analysis of an independently performed IP experiment showed a > 60-fold and statistically highly significant enrichment of the bait *Nt*MatK-HA and of the tobacco orthologs of *At*MKIP1, *At*ValRS2 and *At*EMB3120 (Fig. 3e). In addition, *Nt*RNC1 and a CRM (chloroplast RNA splicing and ribosome maturation) domain protein, both homologs of plastidial splicing factors from maize or Arabidopsis[21,42], were significantly, albeit weakly enriched in these IPs. Several plastidial chaperonins were also enriched (e.g., chaperonins-60 α and β; Supplementary Table 1), possibly indicating a need for folding assistance for *Nt*MatK-HA complex formation. However, *Nt*RNC1, the CRM domain protein and chaperonins were only enriched 2 to 6-fold, suggesting a relatively loose and/or transient association with *Nt*MatK-HA. Furthermore, none of them were enriched in the Arabidopsis IP-MS experiments (Fig. 3a).

To investigate if the MKIP1 interactors together form one complex, we analyzed chloroplast stromal extracts from *Nt*MatK C+ tobacco and wild-type Arabidopsis plants using sucrose density gradient centrifugation and size-exclusion chromatography (SEC). In SEC of *Nt*MatK C+ stromal extracts, *Nt*MatK-HA, *Nt*MKIP1, and *Nt*ValRS2 co-migrated in high-molecular weight fractions (sample fraction ≤8), with some *Nt*MKIP1 and *Nt*ValRS2 also present in lower molecular weight fractions (Fig. 4a). Since the anti-HA antibody cross-reacted with Rubisco large subunit (*Nt*rbcL), among others, we purified *Nt*MatK-HA and its interactors by anti-HA IP from all fractions. This confirmed the presence of *Nt*MatK-HA, *Nt*MKIP1, and *Nt*ValRS2 in high-molecular weight fractions (Fig. 4b). *Nt*EMB3120 could not be detected by immunoblotting. However, proteomic MS/MS analysis of the unpurified SEC fractions 5 and 6, containing the highest amounts of *Nt*MKIP1 and *Nt*MatK, showed that *Nt*EMB3120 was also present in these high-molecular weight fractions (Fig. 4c).

On sucrose gradients, *Nt*MatK-HA, *Nt*MKIP1, and *Nt*ValRS2 again migrated into high-molecular weight fractions (Supplementary Fig. 9a). Similarly, when assessing extracts from Arabidopsis, *At*MKIP1 and *At*ValRS2 co-migrated in high-molecular weight fractions of sucrose gradients (*At*MatK and *At*EMB3120 could not be detected by immunoblotting; Supplementary Fig. 9b). The migration of *At*MKIP1 and *At*ValRS2 did not change upon treatment of extracts with RNase, indicating that the formation of a high-molecular weight MKIP1 complex does not depend on intact RNA (Supplementary Fig. 9b).

To gain insight into the complex' stoichiometry, we tested the subunits for potential self-association. No self-association was observed for *At*MKIP1, *At*ValRS2 and *At*EMB3120 in IP experiments after transient expression in *N. benthamiana* (Supplementary Fig. 10a-c). We were not successful in expressing *At*MatK in this

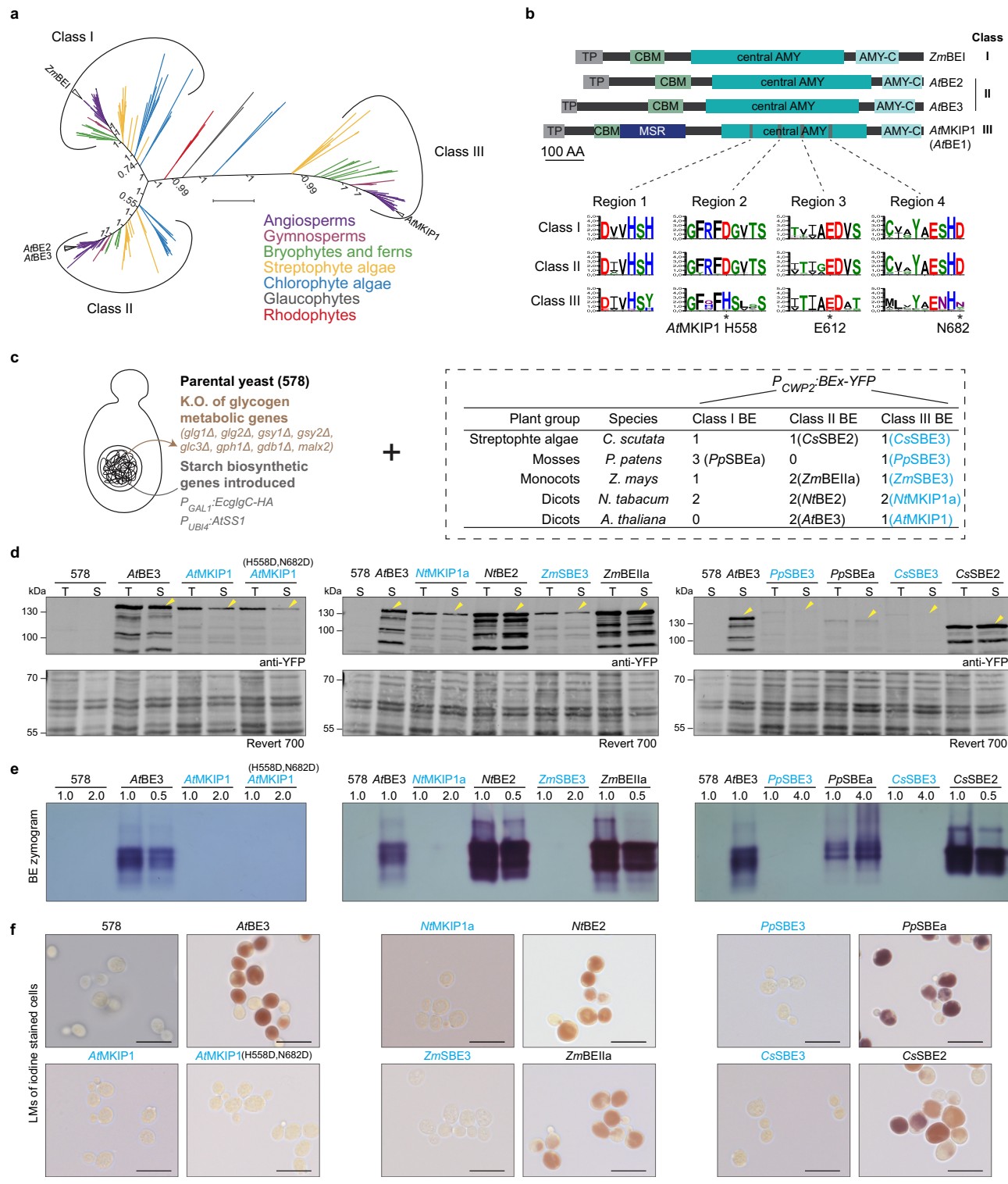

system, nor stably in Arabidopsis, despite using a version codon-optimized for the Arabidopsis nuclear genome. However, we could express *At*MatK with epitope tags as a soluble protein in yeast, allowing us to conduct IP experiments in this system. As for the other subunits, *At*MatK did not efficiently interact with itself, while it did interact with *At*MKIP1 (Supplementary Fig. 10d). Together, our data indicate that the four subunits together form a stable complex. This is consistent with the complex model from AlphaFold3, which predicts a four-subunit heteromultimer of *At*MKIP1, *At*MatK, *At*ValRS2 and *At*EMB3120 with a 1:1:1:1 stoichiometry with intermediate confidence (Fig. 5). AlphaFold3 did not support homo-dimerization of

either component, nor addition of a second copy of any of the four proteins (Supplementary Fig. 10e).

AlphaFold3 further confidently predicted that *At*MKIP1 and *At*MatK directly interact via a large interaction platform formed by the CBM-MSR region of *At*MKIP1 and the N-terminal region of *At*MatK (Fig. 6a–d). To test a direct interaction, we co-expressed *At*MatK-HA with *At*MKIP1-YFP or *At*BE3-YFP as control for IP experiments in yeast. As predicted, the bait *At*MKIP1-YFP efficiently co-precipitated *At*MatK-HA (strain D), whereas *At*BE3-YFP did not (strain C; Fig. 6e). In reciprocal interaction assays, *At*MatK-HA efficiently co-precipitated *At*MKIP1-YFP, but not *At*BE3-YFP (Supplementary Fig. 11b). These results

**Fig. 2 | *At*MKIP1 and orthologous proteins are non-canonical, catalytically inactive branching enzymes (BEs). a** Unrooted Bayesian phylogenetic tree of 254 BE proteins from 85 algal and land plant species (listed in Supplementary Fig. 4a) colored according to their phylogenetic groups. Relevant posterior branch probabilities are indicated next to the branches. Scale bar: 1 amino acid substitution per site. **b** Schematic representation of Arabidopsis BEs from different classes. Maize BEI (*Zm*BEI) is included as a representative of class I BE, which is absent in Arabidopsis. Boxes depict the chloroplast transit peptide (cTP, *gray*), the carbohydrate-binding module (CBM, *green*), the MKIP1-specific region (MSR, *dark blue*), the central α-amylase domain (central AMY, *cyan*), and the C-terminal α-amylase domain (AMY-C, *light cyan*). WebLogos show protein motifs highly conserved in class I and II BEs, based on the sequence alignment used for the phylogenetic tree in (**a**). Asterisks mark the catalytic triad. **c** Genetic modifications in yeast strain "578", enabling it to produce starch-like glucans upon introduction of an active BE. The

box lists the plant BEs expressed in strain "578" together with the numbers of BEs present in the respective species. Class III BEs are highlighted in blue. BEs were stably expressed from the yeast nuclear genome via the strong constitutive *CPW2* promoter and fused to a C-terminal YFP tag. **d** Immunoblot detection of YFP-tagged BEs in total (T) or soluble (S) protein extracts from yeast. Yellow arrows mark the protein bands matching the proteins' molecular weights. Revert 700, total protein stain. **e** BE activity gels (zymograms) of yeast soluble protein extracts. Native gels containing glycogen were incubated in the presence of glycogen phosphorylase, the activity of which is stimulated by the presence of an active BE. Glucan products were visualized by iodine staining of the gel. Numbers indicate the relative protein load. **f** Light micrographs of yeast cells stained with iodine. Three images were taken per line, representative images are shown. Dark brown/purple staining indicates the presence of glucans. Scale bars: 10 μm.

---

confirm a direct and specific interaction between *At*MKIP1 and *At*MatK. We next tested the involvement of the CBM-MSR region of *At*MKIP1 in this interaction (Fig. 6e). An *At*MKIP1-YFP variant without MSR (ΔMSR; strain E) co-precipitated much less *At*MatK-HA than the entire *At*MKIP1-YFP (strain D) in yeast. In turn, MSR-YFP co-precipitated *At*MatK-HA (strain F) only slightly less efficiently than the entire *At*MKIP1-YFP (strain D). When testing an *At*MKIP1 segment comprising its CBM and MSR, little YFP-tagged protein was expressed and recovered in the anti-YFP IP (strain G) for reasons unknown. The precipitated CBM-MSR-YFP protein, however, efficiently co-precipitated *At*MatK-HA. These results indicate a key role of the CBM-MSR region of *At*MKIP1 in interacting with *At*MatK.

Equivalent IP experiments using *At*MatK-HA segments showed that the N-terminal region of *At*MatK-HA co-precipitated well when using either the whole *At*MKIP1 (strain I) and the CBM-MSR region of *At*MKIP1 (strain J) as baits, whereas the C-terminal X domain of *At*MatK-HA did not co-precipitate with *At*MKIP1 (strains M and N) (Fig. 6f, g). Similarly, in reciprocal IPs using *At*MatK-HA as bait, its N-terminal portion but not the C-terminal X domain strongly co-precipitated *At*MKIP1-YFP (Supplementary Fig. 11b). Collectively, these data suggest that the CBM-MSR of *At*MKIP1 and the N-terminal region of *At*MatK form an interaction platform. Contrarily, *At*MatK's X domain likely contributes little or not at all to the protein-protein interactions with *At*MKIP1 and – according to the AlphaFold3 complex prediction – also to the other complex subunits (Fig. 5b).

### *At*MKIP1 is associated with group IIA introns in vivo

We conducted RNA co-IP coupled with RNA sequencing to identify RNA regions specifically associated with *At*MKIP1 or its *At*MatK complex. We therefore immunoprecipitated RNA associated with *At*MKIP1-YFP in Arabidopsis seedlings, using a line expressing *At*BE3-YFP as control. Immunoblot analysis showed the binding of *At*MKIP1's interaction partners, confirming the success of the IP (Fig. 3b). Analysis of the RNA segments that were differentially enriched showed that *At*MKIP1-YFP effectively co-precipitated the RNAs of all group IIA intron containing genes previously identified to be associated with *Nt*MatK-HA[12,13] (Fig. 7a). This enrichment was not due to greater abundance of these RNAs in the *At*MKIP1-YFP expressing line, as they were not or only slightly more abundant ( < 2-fold) in the input RNA fraction of this line. *At*MKIP1-YFP also co-precipitated RNAs of the intron-free genes *rpl23* and *rps19*, which are co-transcribed within the *rpl2*-containing operon[43]. However, while *rpl2* RNA was markedly and significantly enriched throughout the intron and exons, only short segments ( < 200 nt) of *rpl23* and *rps19* flanking *rpl2* were >4-fold enriched, suggesting co-enrichment with *rpl2* (Fig. 7b). The lack of co-enrichment of distal transcript segments is likely due to the nuclease activities present in the chloroplast stroma and during the IP bead incubation step, which lead to RNA fragmentation except for regions protected by proteins. This enables the specific recovery of the RNA regions that are directly or

indirectly (e.g., via a protein or RNA bridge) associated with the bait protein[12,13].

Fine-mapping of the other RNA regions co-precipitated with *At*MKIP1-YFP showed that *trnK* enrichment peaked at the junction between intron domains III and IV, just upstream of the *matK* open-reading frame (ORF) (Fig. 7b). The exons and most other *trnK* intron regions were also enriched, overall largely resembling the enrichment observed by RNA co-IP of *Nt*MatK-HA[12,13]. The remaining binding targets shared a strong enrichment of intron domain II. For *rps12* intron 2, enrichment was low and domain II was the only intron domain significantly enriched, contrasting the pattern observed for *Nt*MatK-HA[13]. However, for most other targets (*atpF, rpl2, trnI*, and *trnA*), strong enrichment was also observed in intron domains IV and VI. In the case of the *trnV* intron, enrichment was relatively low, consistent with its variable enrichment by *Nt*MatK-HA[12,13]. In addition, domain VI of the *trnV* intron was not enriched; this domain may be minimally involved in intron splicing of *trnV* because it lacks the bulged adenosine normally used in the first splicing step[15]. Overall, these results show a remarkable overlap of the intron regions associated with *At*MKIP1-YFP and those previously identified for *Nt*MatK-HA[12,13], suggesting a close functional link between MKIP1 and MatK.

### *At*MKIP1 facilitates splicing of plastidial group IIA introns

The deficiency of *At*MKIP1 beyond the cotyledon stage in the $P_{ABI3}$-rescue plants led to seedling lethality, limiting the material available for splicing analyses. The likely pleiotropic defects of these lines also make identifying the primary defect difficult, since splicing and translation in plastids are linked. Because group IIA introns disrupt tRNAs and transcripts of ribosomal proteins, severe splicing defects will entail defects in translation. In turn, severe plastidial translation defects can reduce the abundance of plastome-encoded MatK, thereby indirectly affecting group IIA intron splicing[44,45]. To catch the primary defect of *At*MKIP1 loss, we inducibly downregulated *At*MKIP1 at later developmental stages using two estradiol-inducible artificial micro-RNA silencing lines in Arabidopsis, each targeting a different site of the *At*MKIP1 mRNA (*amiR-mkip1* plants). Additionally, we created an equivalent line targeting *AtuL4c* mRNA, which encodes an essential protein of the plastidial large ribosomal subunit[27,46]. This *amiR-uL4c* line was used as a control for splicing defects caused by a plastidial translation defect itself.

Estradiol treatment of *amiR-uL4c* plants resulted in pale newly developed leaves, especially at their base, where the tissue is the youngest and chloroplasts are still differentiating (Fig. 8a). Chlorosis was milder in treated *amiR-mkip1* plants but also apparent, and chloroplasts from pale *amiR-mkip1-1* leaves had abnormally few thylakoid membranes (Supplementary Fig. 12). We focused on newly developed leaves (red arrows in Fig. 8a) in downstream molecular analyses. Protein quantification by immunoblotting and proteomics (Fig. 8c, d) showed that *amiR-uL4c* and *amiR-mkip1* tissues were

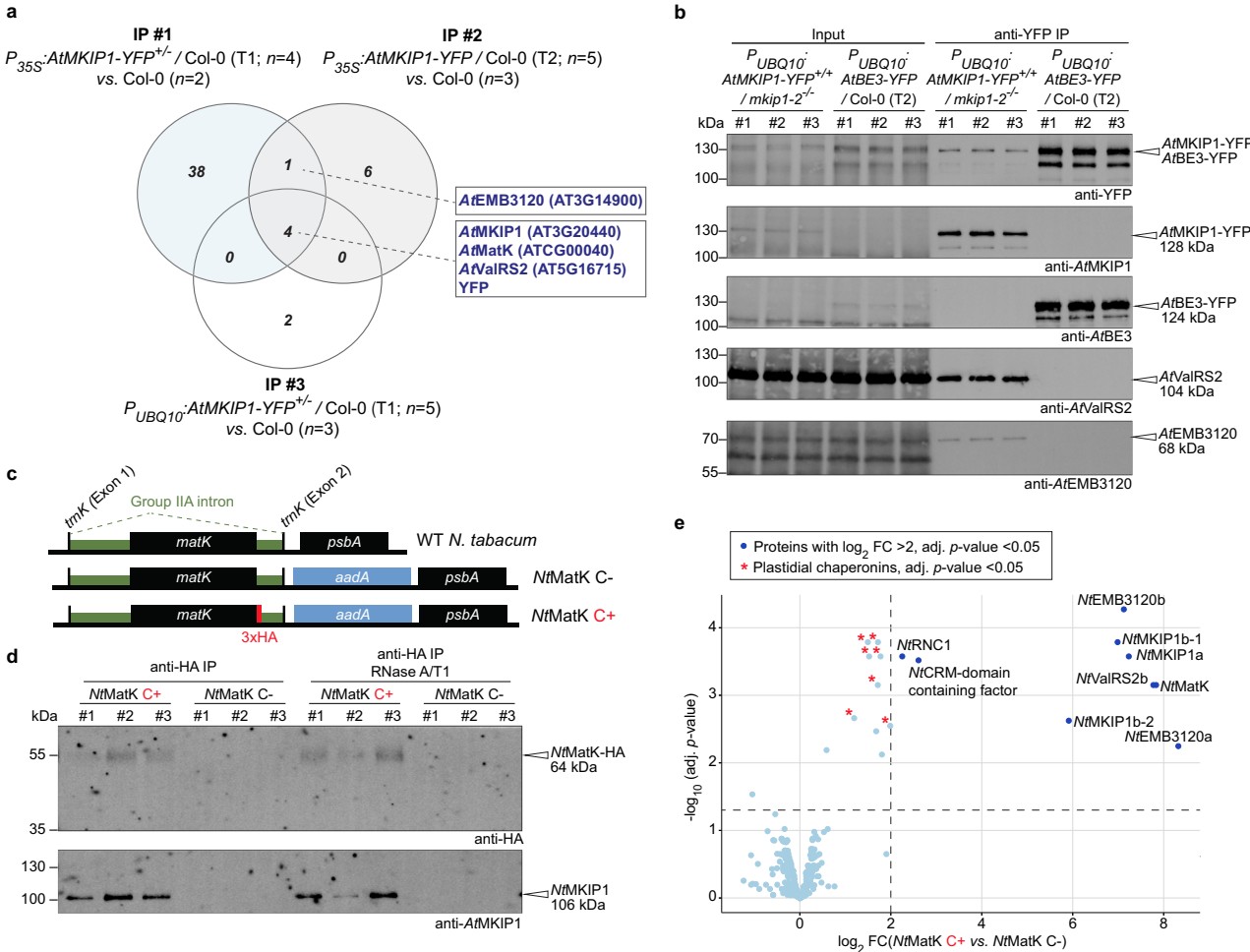

**Fig. 3 | MKIP1 interacts with MatK in Arabidopsis and tobacco in vivo.**
**a** Identification of proteins associated with *At*MKIP1-YFP in mature rosettes of the indicated Arabidopsis lines. Soluble proteins were subjected to anti-YFP immuno-precipitation (IP) coupled to mass spectrometry (MS). The Venn diagram shows the number of proteins significantly enriched (log$_2$ fold change ≥ 2, *p*-value < 0.05) in the *At*MKIP1-YFP expressing lines compared with the wild-type (Col-0) control in three independent experiments. *n*, number of replicate plants. **b** Immunoblot detection of proteins associated with *At*MKIP1-YFP vs. *At*BE3-YFP (control) in 10-day-old Arabidopsis seedlings. Three replicates, each consisting of ~10 g (fresh weight) of pooled 10-day old seedlings, were analyzed per line. Soluble extracts were analyzed before IP (input) and after anti-YFP IP. This experiment was repeated once with equivalent results, except that *At*EMB3120 was not detectably pulled-down by *At*MKIP1-YFP, presumably due to the use of less input material and the limited sensitivity of the *At*EMB3120 antibody (Supplementary Fig. 7). **c** Scheme of the *matK* plastid genome region in WT and transplastomic tobacco lines. *Nt*MatK C + expresses *Nt*MatK with a C-terminal 3xHA tag (*red*). *Nt*MatK C- is a control line

containing the *aadA* marker gene but no *matK* modification. **d** Anti-HA IP experiments using *Nt*MatK-HA as bait. Soluble protein extracted from 10-day-old *Nt*MatK C+ or *Nt*MatK C- (control) seedlings was subjected to anti-HA IP, with or without RNase A/T1 treatment during the incubation with anti-HA beads. *n* = 3 seedling pools. **e** Volcano plot of proteins identified by IP-MS using *Nt*MatK-HA as bait. Anti-HA IP was performed independently from that in (**c**) but using the same workflow without RNase treatment (*n* = 3 seedling pools). Statistical significance was assessed within the Amica platform, employing empirical Bayes-moderated two-sided *t*-tests with Benjamini–Hochberg false-discovery rate correction for multiple testing. Proteins significantly enriched in *Nt*MatK C+ IP (adjusted *p*-value < 0.05) with mean log$_2$ fold change (FC) > 2 are highlighted in dark blue. Significantly enriched proteins annotated as plastidial chaperonins are marked by red asterisks. Multiple isoforms (homoeologs) are observed because tobacco is allotetraploid. All proteins enriched in *Nt*MatK C+ IP with log$_2$ FC > 1 with their Arabidopsis orthologs and spectral counts are listed in Supplementary Table 1.

strongly depleted of the respective proteins targeted by silencing. However, all silencing lines had higher levels of *At*ValRS2 and *At*EMB3120, and estradiol-treated *amiR-uL4c* leaves additionally had higher levels of *At*MKIP1. These alterations are consistent with the putative roles of *At*MKIP1, *At*ValRS2 and *At*EMB3120 in plastid gene expression and the increased expression of such genes when chloroplast differentiation is delayed[47].

We next assessed the newly developed leaves for defects in plastidial translation. In estradiol-treated *amiR-uL4c* leaves, we observed a significant reduction of almost all quantified plastome-encoded proteins, indicating a defect in plastidial translation (Fig. 8b). This tissue was depleted in plastidial ribosomes, as the levels of plas-tidial rRNAs (Fig. 8e, f) and of ribosomal proteins (Supplementary

Fig. 13c) were strongly reduced. We also detected plastidial rRNA processing intermediates (Supplementary Fig. 14); this is consistent with the view that some rRNA processing steps take place after rRNA incorporation into the ribosomal subunit(s), necessitating enough pre-ribosomes and thus plastidial translation for normal rRNA processing[1,48–51].

Defects in plastidial translation, ribosome accumulation and rRNA processing were also apparent upon silencing of *AtMKIP1* but in a considerably milder form (Fig. 8, Supplementary Figs. 13c and 14). Silenced *amiR-mkip1* leaves showed no significant change in *At*MatK abundance (Fig. 8d). Milder pleiotropic defects were also evident when comparing all 445 differentially abundant proteins (compared to the WT) among the 6889 proteins quantified by proteomics, where

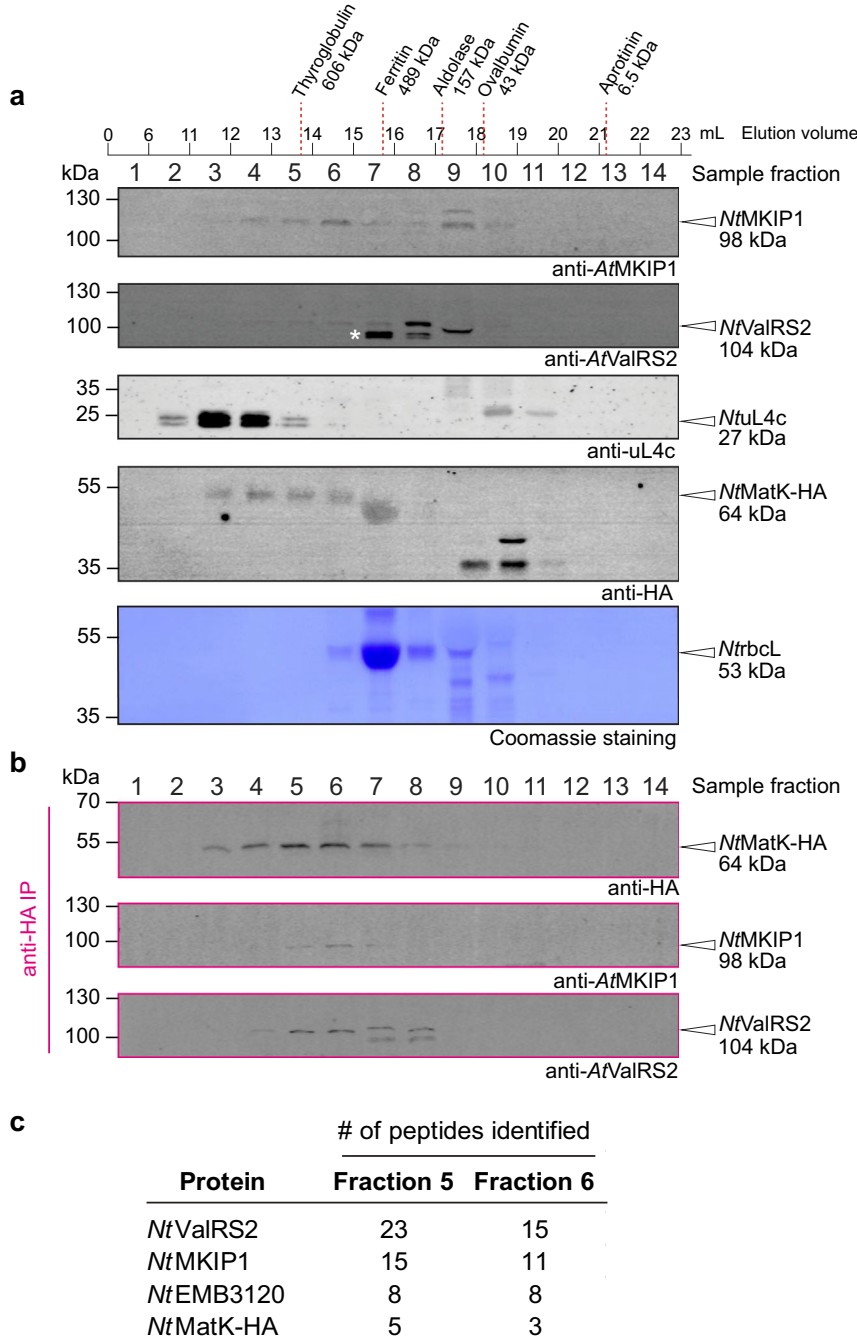

**Fig. 4 | The MKIP1 interactors co-elute in size-exclusion chromatography.**
**a** Chloroplast stroma proteins from *Nt*MatK C+ tobacco leaves were fractionated by size-exclusion chromatography and assessed by immunoblotting. Elution volumes 1-6 ml and 6-11 ml were pooled as sample fractions 1 and 2, respectively. The remaining 1-ml fractions were analyzed separately. The peak elution volumes of globular protein standards fractionated in the same way are indicated. Note that for non-spherical proteins the apparent molecular weights can deviate from those of the markers. The protein indicated by an asterisk may represent a shorter *Nt*ValRS2 version (accession A0A1S4BH67). *Nt*uL4c and *Nt*rbcL are components of the plastidial large ribosomal subunit and of Rubisco, respectively. **b** Anti-HA immunoprecipitation (IP) of the fractions from the experiment shown in (**a**), followed by immunoblotting. **c** MS/MS analysis of the fractions 5 and 6 from the experiment shown in (**a**). Only putative *Nt*MatK complex components are shown. Peptides associated with multiple isoforms (homoeologs) were grouped. In total, 1233 proteins were identified with at least two peptides (**Source Data**).

changes were generally strongest in *amiR-uL4c* and mildest in *amiR-mkip1-2* plants (Supplementary Fig. 13).

We next assessed the efficiency of intron splicing in newly emerged, silenced tissue. We therefore quantified the relative ratios of spliced to unspliced plastid mRNAs using reverse transcription quantitative real-time PCR (RT-qPCR)[52,53]. Given the difficulties associated with reverse transcription of tRNAs, their splicing was determined by northern blotting, circumventing cDNA synthesis. Splicing of the group IIB intron in *rps16* was not assessed, since this is a pseudogene in Arabidopsis with no apparent splicing even in the WT[54,55].

No significant splicing defects compared to WT were observed in mock-treated plants of the *amiR-mkip1* lines (Fig. 9b, Supplementary

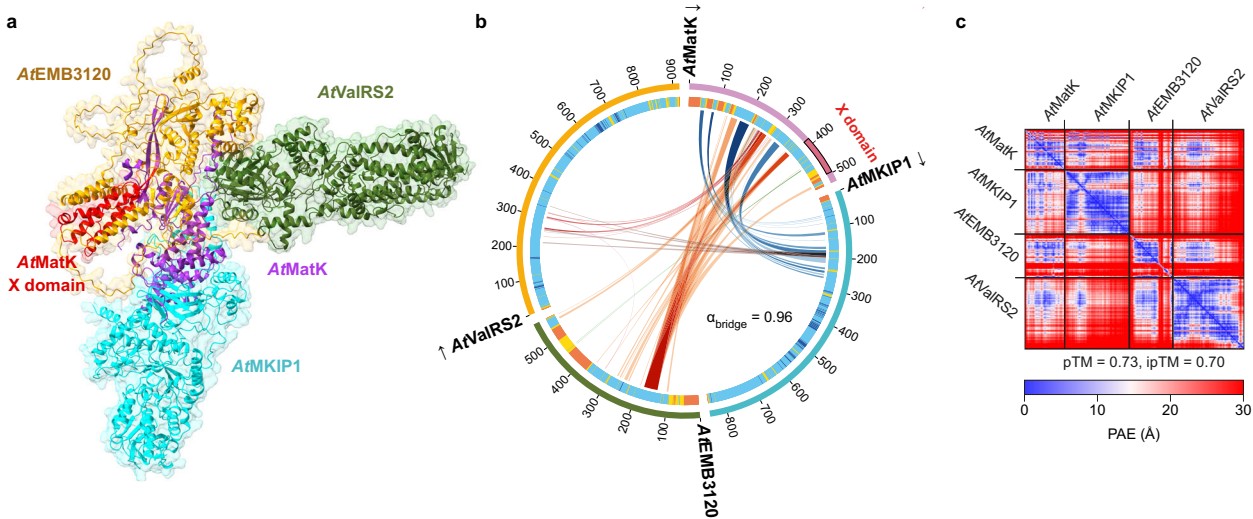

**Fig. 5 | AlphaFold3 predicts the formation of a four-subunit complex comprising *At*MatK, *At*MKIP1, *At*EMB3120 and *At*ValRS2. a** AlphaFold3 structure model of the complex from Arabidopsis. Proteins were modeled without their predicted chloroplast transit peptides, if any. The X domain of *At*MatK is depicted in red. **b** AlphaBridge diagram showing the intermolecular interfaces as bridges between the subunits. Bridges in the same hue belong to the same interface. Interfaces were detected using the default (0.7) confidence level. The inner ring color shows the confidence according to the predicted local distance difference test (pLDDT) (orange, very low; yellow, low; light blue, high; dark blue, very high). *At*MatK's X domain (*red*) begins at the last residue of an interface with *At*EMB3120 and is essentially free of predicted contact sites (in total 6 contact residues). $\alpha_{bridge}$, AlphaBridge score with range 0 (not confident) to 1 (confident). **c** Predicted aligned error (PAE) plot showing the predicted position error between all paired residues. pTM, predicted template modeling score; ipTM score, interface predicted template modeling score.

Figs. 15 and 16). However, upon estradiol-induced silencing, young *amiR-mkip1* tissue showed major splicing defects in most of the introns identified to be associated with *At*MKIP1-YFP. The introns of *trnK*, *trnA*, *rps12* (intron 2), *rpl2*, and *atpF* showed severely reduced splicing efficiency (i.e., a lower ratio of spliced to unspliced RNA) and less absolute spliced RNA compared to estradiol-treated WT and *amiR-uL4c* plants (Fig. 9, Supplementary Fig. 15). Mild splicing defects in *AtMKIP1* silenced tissue were also observed for *trnI* and *trnV* (Fig. 9b, c). In *AtuL4c* silenced tissue, the MatK binding targets showed only little or no impaired splicing (Fig. 9, Supplementary Fig. 15). Despite the slight reduction in *At*MatK (-29 ± 3%; Fig. 8d), its abundance thus seemed still largely sufficient for intron splicing. This is in agreement with a previous report of Arabidopsis that MatK targets are correctly spliced despite impaired plastid translation[29].

The ratio of spliced to unspliced RNA upon *AtMKIP1* silencing was also somewhat reduced in several introns not associated to *At*MKIP1-YFP (Fig. 9e). However, in all cases other than the *petB*, *petD* and *ndhA* intron, this was due to the accumulation of the unspliced precursor, while the spliced RNA appeared not to be reduced compared to WT (Supplementary Fig. 15). In addition, here *AtuL4c* silencing caused analogous defects, suggesting that these are secondary effects associated with chloroplast malfunction[2]. Collectively, these findings strongly suggest that *AtMKIP1* silencing causes a specific and primary splicing defect in all introns associated with MatK and MKIP1. These defects likely underlie the observed plastidial translation defects, ultimately leading to pale leaves in the *amiR-mkip1* plants, and, in the *mkip1*[-/-] lines, to the early embryonic defects.

## Discussion

*At*MKIP1 has been suggested to function in plastid carbohydrate metabolism, largely based on its annotation as a BE and the reduced starch content of a hypomorphic *mkip1* mutant[26,56]. The observation that MKIP1 proteins have lost catalytic activity on glucans[23,25] (Fig. 2) renders such an involvement unlikely. Instead, our work provides several lines of evidence that the essential function of *At*MKIP1 is to assist plastidial splicing in conjunction with *At*MatK. First, MKIP1 physically interacts with MatK, not requiring a protein or RNA bridge (Figs. 3 and 6). Second, RNA-IP experiments demonstrate that *At*MKIP1 is associated with the same intron RNAs as *Nt*MatK[12,13] in vivo (Fig. 7). Third, induced *At*MKIP1 knockdown results in a primary splicing defect specifically in all MatK binding targets (Fig. 9), reinforcing a direct role of *At*MKIP1 in the splicing of *At*MatK targeted introns.

A function in plastid intron splicing helps to explain the drastic consequences on plant viability observed upon *At*MKIP1 loss, since defects in plastid gene expression in Arabidopsis lead to embryonic defects and frequently lethality[39,42]. It is also consistent with the apparent emergence of the *MKIP1* gene within streptophyte algae (Fig. 2), which coincides with the acquisition of the plastidial *matK* gene and group IIA introns[14]. In agreement with this, we identified at least one *MKIP1* gene in all land plant species assessed, except for two non-photoautotrophic parasitic *Cuscuta* species that also lost the *matK* gene and its associated plastid introns[57], possibly rendering MKIP1 dispensable (Supplementary Fig. 4a). In contrast to canonical BEs, MKIP1 has also not been identified in starch-associated proteomes[58,59] but instead in the "core" set of ~200 plastidial nucleoid-associated proteins in maize and Arabidopsis[60,61], together with MatK and other proteins involved in gene expression. *At*MKIP1-YFP did not show the punctate suborganellar distribution pattern typical of nucleoids[62] (Fig. 1b). However, the low signal of $P_{UBQ10}$::*At*MKIP1-YFP necessitated strong $P_{35S}$-driven overexpression, which could have caused (partial) mislocalization if nucleoid association requires complex formation with *At*MatK or other proteins that are lower in abundance.

The neofunctionalization of MKIP1 likely was a multi-step adaptation process. A suggestive feature is its MSR that protrudes from the *At*MKIP1 core. The origin of the MSR is unclear, as we could not find any homologous protein or DNA sequences beyond class III BEs. Nevertheless, it is essential for the function of *At*MKIP1 (Supplementary Fig. 6), probably because it confers – together with the CBM – the interaction with *At*MatK (Fig. 6). The MSR may also interact with RNA; it has conserved lysine and arginine residues at its C-terminal end (Supplementary Fig. 5), which are predicted to be surface exposed in the *At*MKIP1-*At*MatK dimer. Interestingly, the CBM appears to have

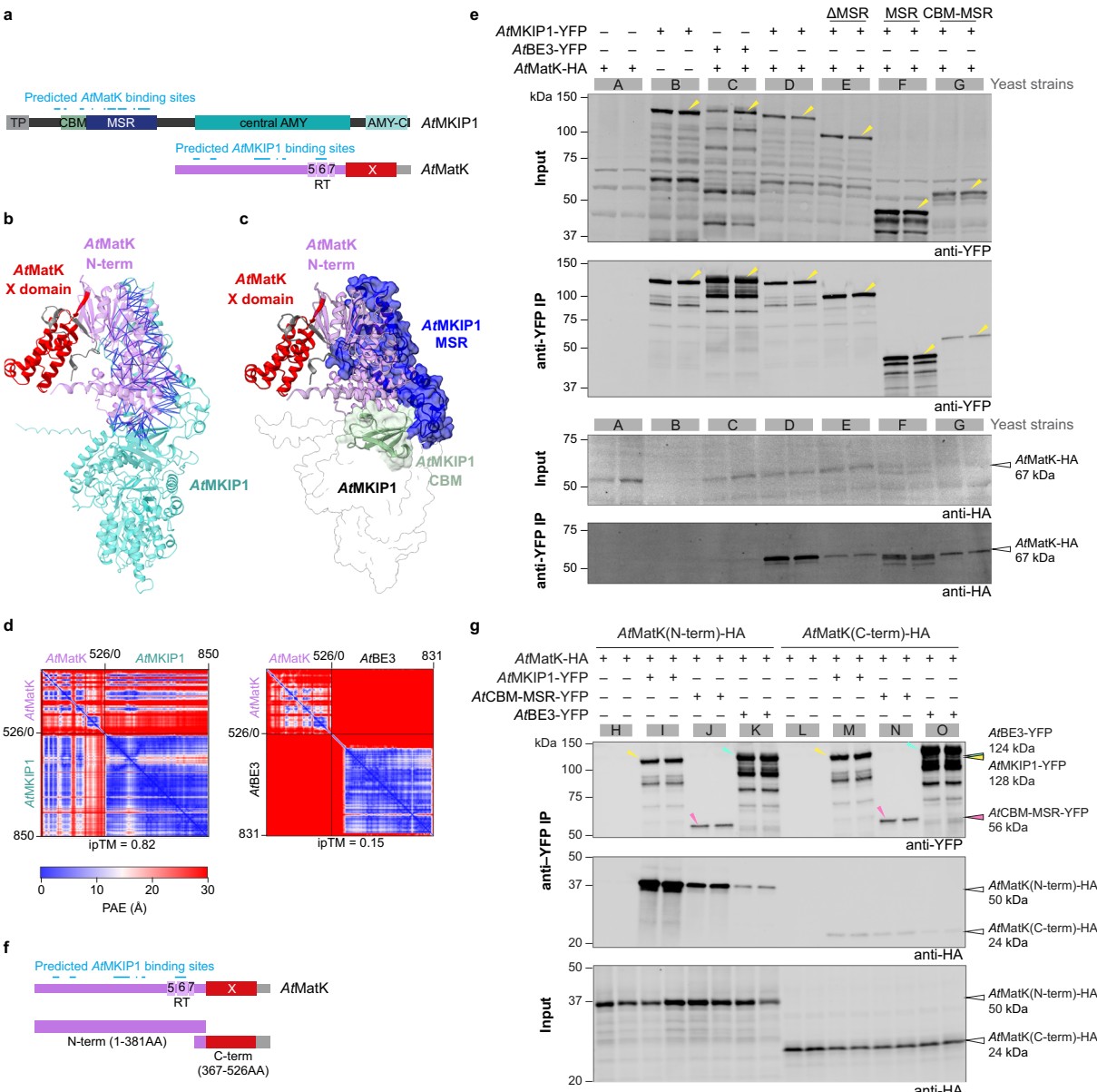

**Fig. 6 | The N-terminal region of *At*MatK forms an interaction platform with the MKIP1-specific region (MSR) and carbohydrate-binding module (CBM) of *At*MKIP1. a** Schematics of *At*MKIP1 (as in Fig. 2b) and *At*MatK. *At*MatK's N-terminal region with its reverse transcriptase (RT) motifs 5-7 (*lilac*) and the X domain (*red*) are indicated. Predicted binding sites (blue lines) are contact sites according to the model shown in (**b**). **b** AlphaFold3 prediction of the heterodimer formed between *At*MatK (colored as in (**a**)) and *At*MKIP1 (cyan). Lines show interprotein residue contacts within 4 Å and a maximum predicted aligned error (PAE) of 5 Å and are colored by their PAE. **c** Model illustrating the domains mediating *At*MatK-*At*MKIP1 dimer formation. *At*MKIP1 is shown as protein surface, highlighting the MSR (*dark blue*) and CBM (*green*). **d** PAE plots of the predicted *At*MatK-*At*MKIP1 and *At*MatK-*At*BE3 heterodimers. ipTM score, interface predicted template modeling score. **e** Immunoprecipitation (IP) of *At*MatK-HA by *At*BE3-YFP (control), *At*MKIP1-YFP or

YFP-tagged *At*MKIP1 variants in soluble protein extracts from yeast. Arabidopsis proteins were expressed without their predicted chloroplast transit peptides, if any. The yeast background was deficient in glycogen to avoid potential interference from BEs binding to glycogen. Two replicate cultures of strains expressing the corresponding protein (+) or not (-) were analyzed. The soluble protein extracts were assessed before IP (input) and after anti-YFP IP using the anti-YFP antibodies. Yellow arrows mark the protein bands matching the molecular weights of the proteins. This experiment was repeated once with similar results. **f** Schematics of *At*MatK (as in a) and its segments used in (**g**). **g** IP experiments performed as in (**e**) but using yeast strains expressing the N- and C-terminal *At*MatK-HA segments shown in (**g**). Anti-YFP input protein abundances are shown in Supplementary Fig. 11c.

adapted to MKIP1's new role. It is unlikely to function in glucan binding, since mutations in the tryptophan residues normally involved in carbohydrate binding to alanine did not impact the function of *At*MKIP1 (Supplementary Fig. 2). Rather, in MKIP1 proteins, the CBM has been repurposed to become part of the MatK-interaction platform, and the *At*MKIP1-*At*MatK dimer model predicts that the CBM is now oriented toward *At*MatK, forming part of the extensive protein-protein interface (Fig. 6). If correct, this would preclude the aforementioned

tryptophans from making contact with glucans. MKIP1 proteins also display extensive sequence divergence across the central and C-terminal α-amylase domains and have lost the integrity of the former catalytic center, rendering them enzymatically inactive as a BE (Fig. 2). These changes may have contributed to the elimination of glucan interaction sites on the protein surface[63], further diverging the function of MKIP1 from that of canonical BEs. However, the α-amylase domains are likely to support MKIP1 functions in other, as yet unknown

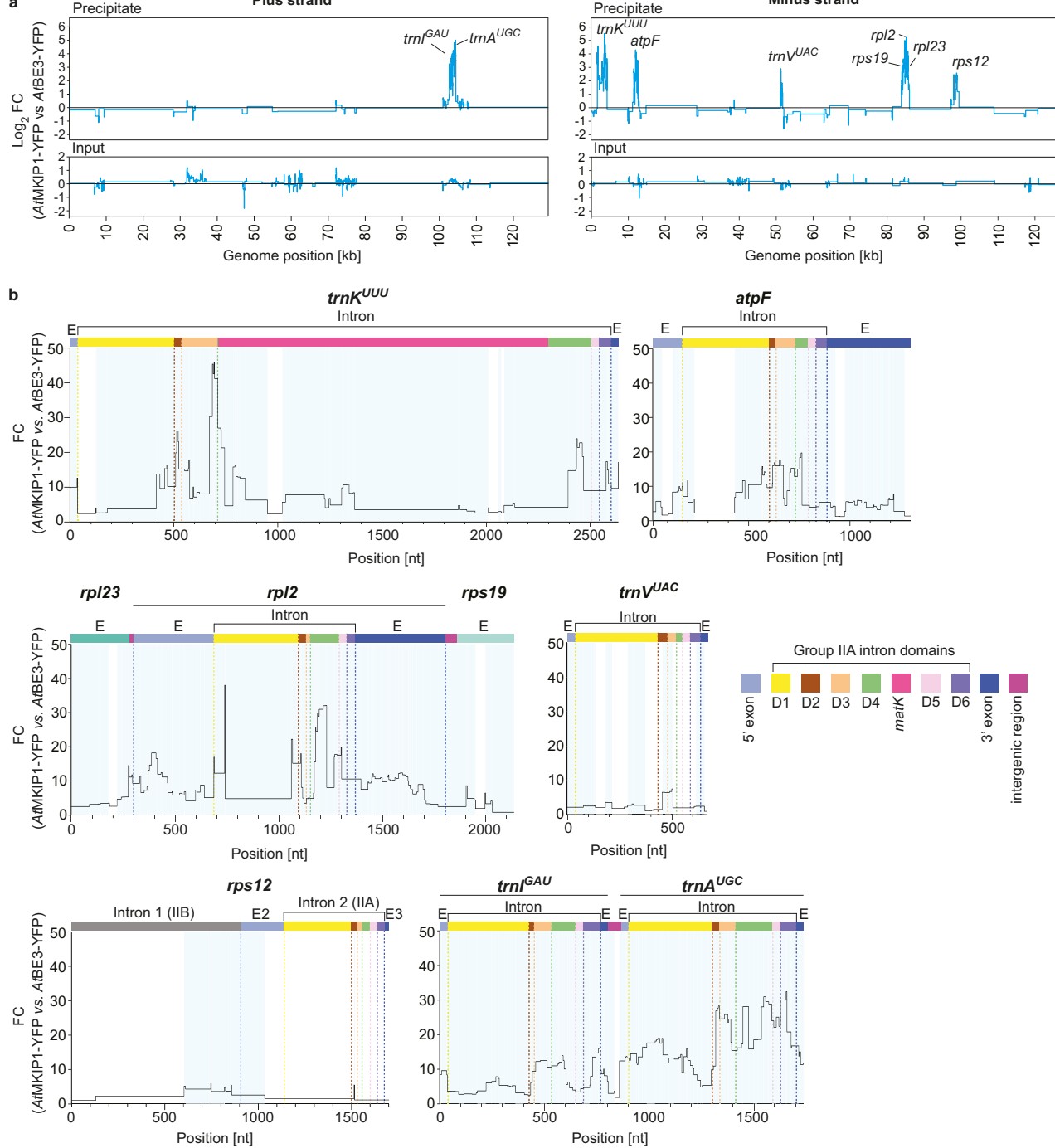

**Fig. 7 | AtMKIP1-YFP co-precipitates plastidial intron RNA. a** Sequencing analysis of the RNA recovered in the precipitates of the anti-YFP immunoprecipitations shown in Fig. 3b and in the input RNA. Three pools of 10-day old Arabidopsis seedlings ($P_{UBQ10}$:AtMKIP1-YFP / mkip1-2$^{-/-}$ and $P_{UBQ10}$:AtBE3-YFP / Col-0) were analyzed per line. $Log_2$ fold changes (FC) in the precipitates indicate the mean relative abundances of the RNA segments co-precipitated by AtMKIP1-YFP compared to those co-precipitated by AtBE3-YFP. $Log_2$ FC in the inputs indicate the mean relative abundance of RNA segments present in the input samples. Relative abundances and statistical significances were calculated by DiffSegR, which combines RNA segments with similar fold changes into regions indicated by horizontal lines, then employs two-sided differential expression testing based on negative binomial generalized linear models and Benjamini–Hochberg false discovery rate correction for multiple testing. Regions were mapped to the Arabidopsis plastome in a strand-specific manner, excluding the second copy of the inverted repeat region. Genes containing regions with a $log_2$ FC > 2 are labeled by name. **b** Zoom-in view of differentially abundant regions ($log_2$ FC > 2) in the precipitates plotted in linear scale. Exon (E) and intron domains are annotated with color-coded boxes, with dashed lines indicating their borders. Brackets indicate group IIA introns. Significantly enriched (adjusted $p$-value < 0.01) regions are highlighted with a light-blue background. Only one half of the group IIB rps12 intron 1 is shown, as it is dipartite (trans-spliced) and the second part was not enriched.

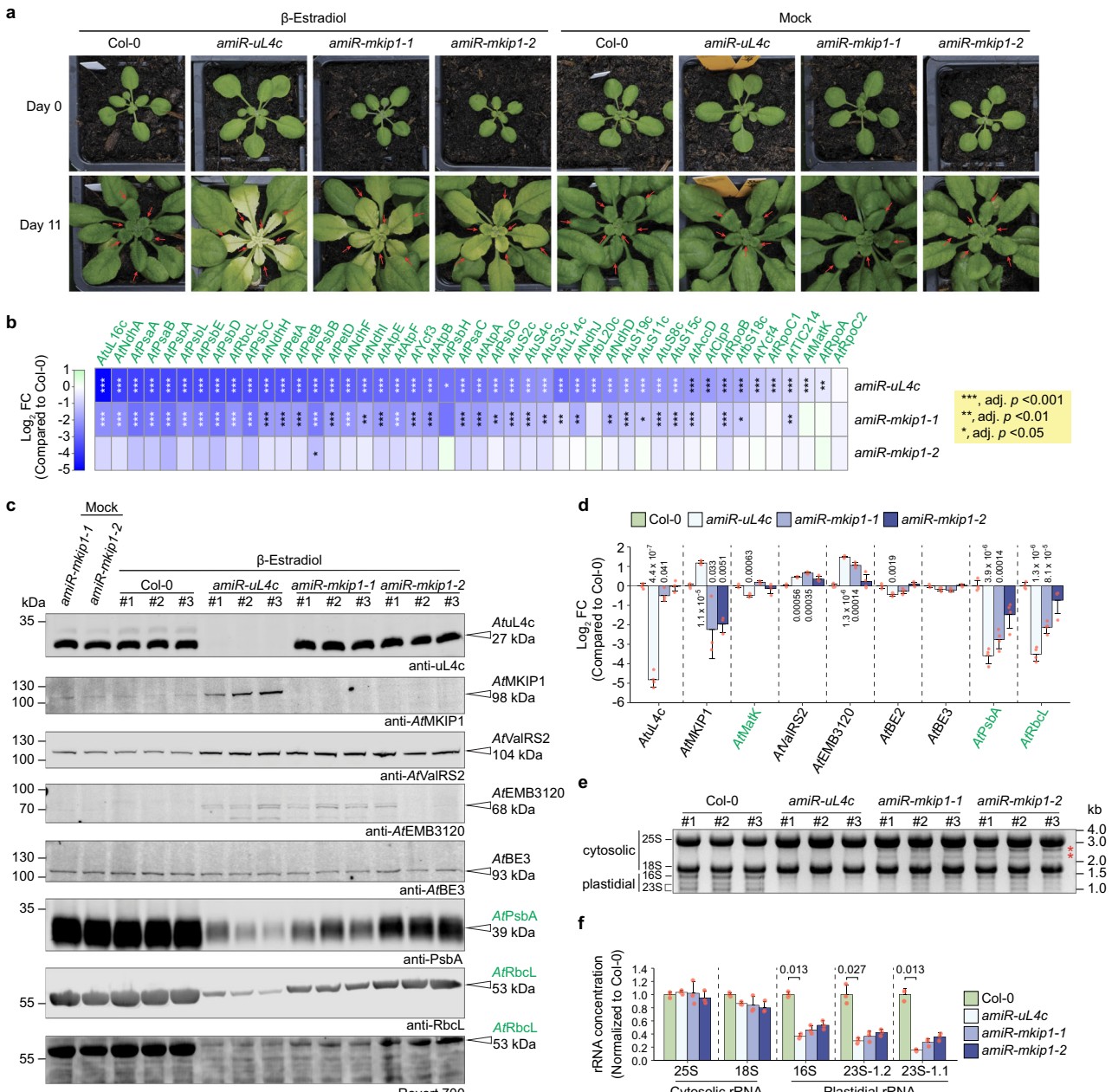

**Fig. 8 | Induced silencing of *AtMKIP1* in Arabidopsis results in pale new leaves with milder plastidial translation defects than *AtuL4c* silencing.** Col-0, wild type. Proteins in green are encoded in the plastome. **a** Arabidopsis rosettes before (day 0;) and 11 days after daily spraying with β-estradiol or mock solution. Red arrows indicate the newly emerged leaves sampled for downstream analyses. Similar phenotypes were observed in 15 replicate plants per line and in two independent repeats of the experiment. **b** Abundances of plastome-encoded proteins in newly emerged silenced tissue quantified by proteomics of total proteins. Shown are mean log₂ fold changes (FC) in the indicated *amiR* lines compared to Col-0 (*n* = 4 estradiol-treated plants). Statistical significance was assessed within the Amica platform, employing empirical Bayes-moderated two-sided *t*-tests with Benjamini–Hochberg false-discovery rate correction for multiple testing. Asterisks indicate adjusted *p*-values compared to Col-0 and are in white if log₂ FC are below -2. **c** Immunoblot analysis of protein accumulation. Total protein from newly emerged leaves after 11 days of treatment was loaded on an equal leaf area basis. Three estradiol-treated plants were analyzed per line. Revert 700, total protein stain. **d** Abundances of select proteins quantified and statistically assessed as in (**b**).

Data are presented as mean log₂ fold changes (FC) compared to Col-0 ± SD with *n* = 4 estradiol-treated plants, except for *At*MKIP1 in *amiR-mkip1-1* and *amiR-mkip1-2*, where *n* = 3 estradiol-treated plants, as it was not detected in one replicate of these lines. Red dots represent individual data points. For significantly different means compared to Col-0 (adjusted *p*-values < 0.05), the adjusted *p*-values are indicated. All *p*-values are provided in the Source Data. **e** Denaturing gel electrophoresis assessing ribosomal RNA (rRNA) accumulation. Of the 23S rRNA, the 1.2 kb and 1.1 kb fragments are indicated. Equal amounts of total RNA were loaded for each sample. *n* = 3 estradiol-treated plants. Asterisks mark two additional bands derived from incomplete 23S rRNA processing. **f** Quantification of rRNA abundance upon β-estradiol treatment using a TapeStation device. Shown are means normalized to Col-0 ± SD (*n* = 3 estradiol-treated plants). Red dots represent individual data points. Statistical significance was evaluated using the Kruskal-Wallis test followed by a two-sided Dunn's post hoc multiple comparisons test and Bonferroni *p*-value adjustment. For statistically significant differences between means (adjusted *p* < 0.05), the adjusted *p*-values are provided above the brackets.

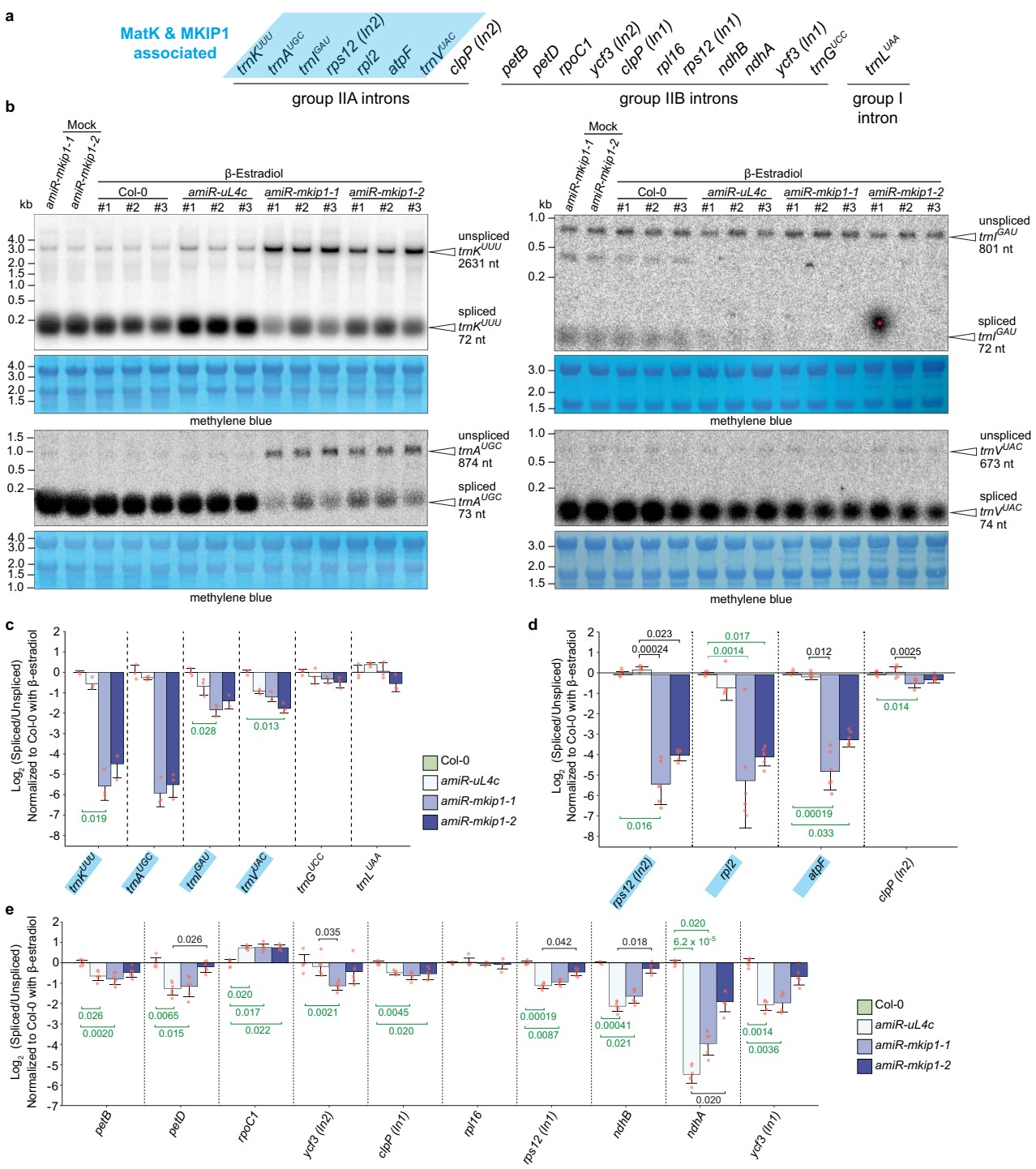

**b**

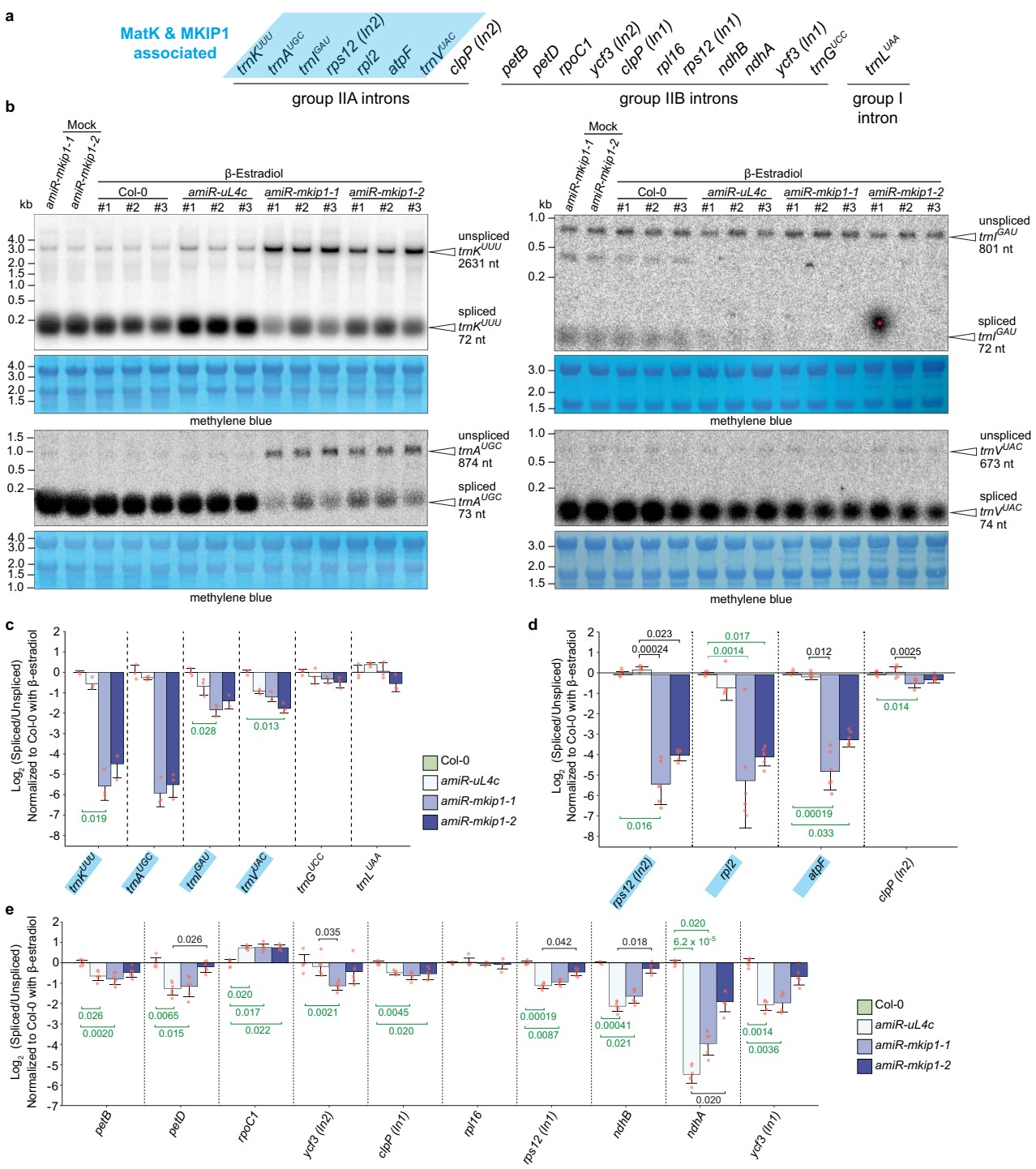

ways, as they are under evolutionary constraint and a P749S substitution in the central α-amylase domain of *At*MKIP1 resulted in pale leaves[56].

MKIP1 is analogous to other enzyme(-like) factors recruited to facilitate plastidial intron splicing, but which no longer require or possess enzymatic activity, including the pseudouridine synthase homolog Raa2, implicated in group IIA trans-splicing in *Chlamydomonas*[64], and maize CHLOROPLAST RNA SPLICING2 (CRS2), a catalytically-dead peptidyl-tRNA hydrolase required for group IIB intron splicing[65]. There are also several nuclear-encoded factors that assist in the splicing of MatK-associated group IIA introns[1,2], some of which possess protein domains originally associated with other

functions (e.g., RNC1 and CRS1 from maize[20,21]). However, none of these factors were reported to be in direct contact with MatK, consistent with their weak or absent enrichment in the *Nt*MatK-HA IP (Fig. 3e), nor do they associate with the very same introns as MatK. This is consistent with upstream roles in intron folding and/or RNA stabilization.

By contrast, the physical and functional association of MKIP1 with MatK suggests a role of MKIP1 in directly helping MatK function. There is currently no experimental structure available for a plastidial group II intron, and we were unable to realistically model one or its association with the MatK complex with AlphaFold3. Nevertheless, MatK itself can likely associate with introns to some extent, as indicated by an early

**Fig. 9 | Silencing of *AtMKIP1* leads to splicing defects in MKIP1/MatK-associated introns.** β-estradiol or mock treatment were performed as described for Fig. 8. In all cases, newly emerged leaves were assessed. An independently repeated analysis of intron splicing in estradiol and mock-treated *amiR-mkip1-1* and estradiol-treated *amiR-uL4c* and Col-0 plants gave similar results. Col-0, wild type. **a** Scheme of the introns present in the Arabidopsis and tobacco chloroplast genome. MKIP1/MatK-associated introns are highlighted in blue. **b** Northern blot analysis of intron splicing in tRNAs using probes against exons 1 of *trnK^{UUU}*, *trnA^{UGC}*, *trnI^{GAU}*, or *trnV^{UAC}*. Methylene blue is a total RNA stain used as a loading control. The red asterisk marks a non-specific contamination. *n* = 3 estradiol-treated plants and *n* = 1 mock-treated plant. **c** Intron splicing efficiency of tRNAs of estradiol-treated plants. MKIP1/MatK associated RNAs are highlighted in blue. Efficiencies are the log$_2$ transformed ratios of spliced to unspliced transcripts of a given sample normalized to that of the estradiol-treated Col-0 mean, based on the data presented in (**b**) and

Supplementary Fig. 16. Shown are means ± SD (*n* = 3 plants), except for *trnI*, where one replicate with non-specific contamination (marked with a red asterisk in **b**) in *amiR-mkip1-2* was excluded. Red dots indicate single data points. Statistical significance was evaluated using the Kruskal-Wallis test followed by a two-sided Dunn's post hoc multiple comparisons test and Bonferroni *p* value adjustment. *trnI* in *amiR-mkip1-2* was excluded from statistical analysis due to insufficient *n*. For statistically significant differences between means (adjusted *p* < 0.05), the adjusted *p*-values are provided, with green values indicating comparisons with Col-0. **d**, **e** Intron splicing efficiency of mRNAs in estradiol-treated plants assessed by RT-qPCR. Splicing efficiency and statistical significance were calculated as in (**c**). Shown are means ± SD (*n* = 6 plants). For statistically significant differences between means (adjusted *p* < 0.05), the adjusted *p*-values are provided, with green values indicating comparisons with Col-0. Underlying abundances of spliced and unspliced mRNAs are presented in Supplementary Fig. 15.

RNA-binding study[66] and the presence of positively charged residues in MatK, in particular in its X domain[18]. MatK's N-terminus may also contribute to RNA binding. However, given its sequence divergence and engagement in interacting with MKIP1 and probably also with EMB3120 and ValRS2, it seems improbable that this region by itself can establish the high-affinity contacts that bacterial maturases use for the initial intron recognition[7,8,17]. The interaction with MKIP1 does not appear to be necessary for MatK stability, as MatK abundance was not significantly affected by *AtMKIP1* silencing (Fig. 8d). Instead, complex formation may equip MatK with a larger area for RNA binding, which may enable it to recognize multiple introns via diverse contact sites, and thus help it to act as a general splicing factor. At the same time, our data and the AlphaFold3 predictions imply that MatK's X domain is barely engaged in protein-protein interactions (Figs. 5 and 6). This likely enables the X domain to interact with the intron's catalytic core and ultimately facilitate splicing, in a manner analogous to the role of this domain of bacterial maturases[10,11].

RNA recognition, binding and splicing may also be assisted by the other MatK complex components ValRS2 and EMB3120. If ValRS2 does indeed play a role in MatK-mediated splicing, it probably has retained its function as a valyl-tRNA synthetase, as it is the only plastid-localized protein with such predicted function[39]. This is consistent with our observation that most of ValRS2 migrates into low-molecular weight fractions (Fig. 4, Supplementary Fig. 9), probably reflecting its monomeric tRNA synthetase activity. Its additional recruitment as a splicing factor would resemble that of bifunctional tRNA synthetases in *Neurospora*[67] and yeast[68], which help the splicing mitochondrial group I introns in addition to having tRNA synthetase function. While the intron associations of *At*MKIP1 and splicing impairments upon *AtMKIP1* silencing suggest that MKIP1 is part of the MatK complex for all of its targets, EMB3120 and ValRS2 might only associate during the splicing of some introns. Alternatively, the composition of the MatK splicing complex could vary during the sequential steps of splicing. The latter situation would resemble the major type of nuclear spliceosomes: these also thought to derive from group II introns and undergo complex remodeling during each splicing step[3]. Further work will be required to dissect the (dynamic) contribution of each component of the MatK complex during intron binding, folding and splicing.

## Methods
### Mutant lines and generation of transgenic plants
Unless indicated otherwise in the figures, experiments were performed on Arabidopsis single-insertion lines of the T3 or a later generation. Where zygosity is not indicated in the Figures, plants were either homo- or heterozygous for the insert. Binary plant expression constructs were transformed into *Agrobacterium tumefaciens* (strain GV3101). *Arabidopsis thaliana* (Col-0) plants were transformed by floral dipping. Transformed T1 plants were selected for their resistance to hygromycin or Basta on 0.8% (w/v) agar plates containing one-half-

strength Murashige and Skoog (MS) medium including vitamins (Duchefa Biochemie) at pH 5.7 with 25 mg l$^{-1}$ hygromycin B or 15 mg l$^{-1}$ glufosinate ammonium (BASTA; from Merck) and further tested for transgene expression by immunoblotting. Single insertion lines were identified based on the segregation ratio of resistant to non-resistant plants in the T2 generation.

T-DNA insertion lines were in Col-0 background from the SALK collection and were obtained from the European Arabidopsis Stock Centre under the following identifiers: SALKseq_040659 (*mkip1-1*), SALK_129083 (*mkip1-2* or *emb2729-2*) and SALK_070161 (*valrs2-1*). Primers used for genotyping are listed in Supplementary Data 1. Insertion sites were confirmed by Sanger sequencing of the PCR amplicon containing the border between the T-DNA and its genomic insertion site.

Homoplastomic *Nicotiana tabacum* (tobacco) lines modified to express *Nt*MatK fused to a C-terminal 3x HA tag (line *Nt*MatK C + ) or control lines carrying only the selection marker (*Nt*MatK C-)[12] were kindly provided by Christian Schmitz-Linneweber (HU Berlin, Germany).

### Cloning of plant constructs
Unless otherwise noted, all plasmids for protein expression in Arabidopsis were generated using Gateway recombination cloning technology (Invitrogen). The plasmids were validated by sequencing of the insert(s) (when PCR-amplified fragments were cloned) and/or diagnostic restriction digests (when PCR was not used).

For *AtMKIP1*, the CDS of splice form 2 (AT3G20440.2) was considered the primary splice form based on the mapping coverage of available RNA-sequencing data at J Browse[69]. The CDS of the full-length primary splice forms of *AtMKIP1* (AT3G20440.2), *AtEMB3120* (AT3G14900.1), and *AtValRS2* (AT5G16715.1) flanked by attB sites were amplified by PCR from Arabidopsis Col-0 cDNA and individually recombined into pDONR221. The resulting vectors (AtMKIP1_pDONR221, AtEMB3120_pDONR221, and AtValRS2_pDONR221) were separately recombined with pUBC-YFP[70], which allows $P_{UBQ10}$-driven expression of proteins carrying a C-terminal YFP tag. For $P_{35S}$-driven expression of *At*MKIP1 with a C-terminal YFP tag, AtMKIP1_pDONR221 was recombined with the gateway vector pB7WG2.

For expression of *At*MKIP1's CDS with a C-terminal mCitrine tag driven by $P_{ABI3}$ or $P_{UBQ10}$, we first amplified the respective promoter sequences (for $P_{ABI3}$: 2.2 kb preceding the start codon of *ABI3*, AT3G24650; for $P_{UBQ10}$: the 0.6 kb promoter as in pUBC-YFP) flanked by attB sites by PCR using Arabidopsis Col-0 genomic DNA or the pUBC-YFP vector as templates. After recombining the PCR products into pDONR P4-P1R, the resulting promoter vectors (PABI3_pDONRP4-P1R and PUBQ10_pDONRP4-P1R) were separately recombined with mCitrine_pDONR P2R-P3 (ref. 38) and AtMKIP1_pDONR221 into the gateway multi-site destination vector pB7m34GW2.

For expression of the CDS of $At$MKIP1, $At$ValRS2, and $At$EMB3120 with a C-terminal 3x FLAG tag driven by $P_{UBQIO}$, the vectors AtMKIP1_pDONR221, AtEMB3120_pDONR221, and AtValRS2_pDONR221 were individually recombined with PUBQ10_pDONRP4-P1R and 3xFLAG_pDONR P2R-P3 (containing a 3x FLAG tag, followed by a stop codon) into pB7m34GW2.

Point mutations in the CBM of $At$MKIP1 (W151A, W171A) or the central α-amylase domain of $At$MKIP1 (E612A, H681A) were introduced into AtMKIP1_pDONR221 using two sequential rounds of site-directed mutagenesis for each substitution (QuikChange; Agilent Technologies). The resulting AtMKIP1(W151A,W171A)_pDONR221 and AtMKIP1(E612A,H681A)_pDONR221 vectors were then separately recombined with PUBQ10_pDONRP4-P1R and mCitrine_pDONR P2R-P3 (ref. 38) into pB7m34GW2. To express a ΔMSR version of $At$MKIP1, the MSR (AA 178-333 of $At$MKIP1) in AtMKIP1_pUBC-YFP was deleted using NEBuilder HiFi DNA Assembly (New England Biolabs) by fusing PCR-amplified vector fragments covering the entire vector except for the MSR, resulting in pBP605_AtMKIP1_deltaMSR(178-333)_pUBC-YFP.

amiRNAs (21mers; Supplementary Data 1) for inducible silencing of $AtMKIP1$ or $AtuL4c$ were designed using the WDM3 MicroRNA Designer tool. Synthesized amiRs in the $MIR319a$ backbone flanked by attB sites were first recombined into pDONR221 and then into the binary pH7m34GW vector[71] containing an estrogen-inducible XVE cassette driven by $P_{UBQIO}$.

## Plant growth and β-estradiol treatment

Arabidopsis and tobacco plants were grown in Percival AR-95L chambers (CLF Plant Climatics) under 12-h light / 12-h dark cycles as previously described[72]. For β-estradiol-induced silencing, 3-week old plants were sprayed once per day with aqueous β-estradiol solution (20 μM β-estradiol [Sigma-Aldrich], 0.2% [v/v] dimethyl sulfoxide [DMSO], 0.01% [v/v] Silwet L-77). The mock spray solution was prepared with the same composition but omitting β-estradiol. Newly emerged leaves after 11 days of treatment were flash-frozen in liquid $N_2$ and stored at -80 °C until further processing. Each replicate derives from an individual plant.

## Proteome analysis of β-estradiol treated plants

For total proteome analyses, frozen newly emerged leaves from Arabidopsis rosettes after 11 days after daily spraying with β-estradiol (see section "Plant growth and β-estradiol treatment") were ground by mortar and pestle in liquid $N_2$. Four biological replicates, each derived from an individual plant, were analyzed per line, using β-estradiol-treated WT (Col-0) plants as control (total number of samples = 16). Some samples were re-run on the MS instrument to optimize the amount of peptides injected, but if so, only the newest run was used for downstream data analysis (i.e., $n = 1$ technical replicate).

About 15 mg of ground plant material per sample was subjected to peptide preparation using the iST kit for plant tissues (PreOmics, Germany) following the manufacturer's instructions. Dried peptides were re-solubilized by sonication in 3% (v/v) acetonitrile, 0.1% (v/v) formic acid. Peptides were spiked with iRT peptides (Biognosys) and analyzed on Orbitrap Exploris 480 mass spectrometer (Thermo Fisher Scientific) equipped with a Nanospray Flex Ion Source (Thermo Fisher Scientific) and coupled to an M-Class UPLC (Waters) equipped with a nanoEase M/Z Symmetry C18 trap column (100 Å, 5 μm, 180 μm x 20 mm, Waters) and a nanoEase M/Z HSS C18 T3 analytical column (100 Å, 1.8 μm, 75 μm x 250 mm, Waters).

Peptides were separated at a flow rate of 300 nl min⁻¹ by the following 90-min gradient: initial hold at 5% solvent B (0.1% formic acid in acetonitrile) / 95% solvent A (0.1% formic acid in water) for 3 min, followed by a gradient to 25% B / 75% A in 80 min, and to 32% B / 68% A in 10 min, a cleaning step at 95% B for 10 min and re-equilibration at 5% B / 95% A for 10 min (all percentages v/v). MS data were acquired in data-independent acquisition (DIA) mode using the following settings:

The full MS was scanned at a resolution of 60,000 with a normalized automatic gain control target value of 300% and a maximum injection time of 45 ms. The mass range was set to 400–1000 m/z. Peptides were fragmented at a normalized collision energy of high-energy collision dissociation of 30%. The normalized automatic gain control target value for fragment spectra was set to 1000%. At the MS2 level, 60 windows of 10 m/z with optimized window placement and without overlap within the mass range of 400–1000 m/z were analyzed at a resolution of 30,000. Maximum injection time mode was set to automatic.

In addition, two pools composed of one biological replicate of each line were separately analyzed by data-dependent acquisition (DDA) for subsequent spectral library built, using the same gradient as for DIA and the following settings: full MS scan with a resolution of 120,000 with a normalized automatic gain control target of 300%, a maximum injection time of 45 ms, and a mass range of 350-1200 m/z. Precursors were filtered for charge states between 2 and 7, selected for MS/MS based on a minimum intensity of 5000 and fragmented by high-energy collision dissociation using a normalized collision energy of 30%. Fragment spectra acquired with a resolution of 15,000. Precursor masses previously selected for MS/MS measurement were excluded from further selection for 20 s, and the exclusion window was set to 10 ppm.

DIA-MS raw files were converted into mzML files using MSConvert[73] (v3.0.2472-63d00b1). The MS data were further processed using MSFragger-DIA implemented in the computational platform FragPipe[74] (v23.1) with DIA-NN[75] (v2.1.0). A hybrid spectral library was built from the primary DIA data and the auxiliary DDA data using the DIA_SpecLib_Quant workflow with default settings. Spectra were searched with MSFragger[76] (v4.3) against the Araport 11 proteome database concatenated to its reversed decoyed fasta database and common protein contaminants using default settings, i.e., allowing for one missed tryptic cleavage and setting carbamidomethylation of cysteine as fixed modification and methionine oxidation and acetylation of the protein N-terminus after methionine removal as variable modifications. DIA data was analyzed with the hybrid spectral library as input and using the same FragPipe workflow and the same search settings against the Araport 11 database as above. Average $\log_2$ fold changes (FC) and adjusted $p$-values were calculated in Amica[77] (v3.0.1) webserver using the DIA workflow, filtering for proteins with at least 3 valid values in each group, setting re-normalization (in addition to that previously conducted by DIA-NN) and imputation of missing values to "none", and differential expression analysis using the implemented $limma$[78] framework. Only proteins that were identified by at least two peptides, with at least one of them being uniquely assigned, were kept for further data analysis. In the three cases where two individual splice-form variants had been quantified separately due to the detection of splice-form specific peptides, the variant with fewer quantified peptides was excluded from the analysis.

For proteins that were strongly and statistically significantly changed in at least one line compared to the wild-type control (i.e., $\log_2$ FC $\leq -1$ or $\geq 1$, adjusted $p$-value < 0.05), the number of appropriate clusters was identified by the Elbow method using the "wss" function, followed by grouping the proteins into 4 clusters by $k$-means clustering using the "kmeans" function in RStudio (v2024.12.1). Corresponding genes that are overrepresented in each cluster compared to those corresponding to the in total 6,688 proteins quantified by proteomics were identified by PANTHER (v19.0, released 20240807) over-representation tests[79] using Fisher as test type, Bonferroni method for $p$-values corrected for multiple comparison, and the GO database released 20250316 (https://doi.org/10.5281/zenodo.15066566) for gene annotation. Protein locations are the predicted/annotated consensus locations retrieved from SUBA5 (ref. 80). Localization enrichments in each cluster were calculated compared to the localizations of the total 6688 proteins quantified by proteomics. In case of multiple

locations, only the first one was considered, since multiple locations in the four protein clusters was deemed negligible (concerning ~1% of the proteins). Plastidial ribosomal proteins were annotated following the nomenclature from Scarpin et al.[81].

## Plant imaging

For imaging Arabidopsis chloroplasts by transmission electron microscopy (TEM), leaf sections were fixed, embedded, and imaged as described previously[72].

For observing *At*MKIP1-YFP localization and co-localisation with either mitochondria or nuclei, rosette leaves from 4-week-old, soil-grown Arabidopsis plants were syringe-infiltrated with infiltration medium (10 mM MES-KOH, pH 5.6, 10 mM MgCl$_2$) containing 100 nM MitoTracker Red CMXRos (from a 1 mM stock in 100% DMSO) or 10 µg ml$^{-1}$ DAPI (dissolved in water). The abaxial leaf sides were imaged using a Zeiss LSM880 confocal microscope equipped with 40× water objective. Excitation was performed with the 405-nm diode (for DAPI), the 514-nm Argon laser (for YFP), the 561-nm Argon laser for Mito-Tracker, and the 633-nm helium-neon laser (chlorophyll). The emitted signals were detected at 410–511 nm for DAPI, 519–566 nm for YFP, 578–643 nm for MitoTracker, and 647–721 nm for chlorophyll. Image acquisition and processing were performed using ZEN Microscopy Software.

To image Arabidopsis embryos, seeds dissected from a silique were collected in a microcentrifuge tube and incubated in a 1:1 mixture of ethanol and acetic acid for 1 h at 20 °C with rotation, then placed on a microscope glass slide and further cleared by incubation with Hoyer's medium (7.5 g gum arabic, 100 g chloral hydrate, 5 ml glycerol and 30 ml water) for 1 h. Cleared seeds were imaged using a Leica DM2500 microscope.

## Plant protein extraction and immunoblotting

Total plant protein extraction from leaf disks and immunoblotting using IRDye 800CW goat anti-rabbit IgG secondary antibody (LICOR-bio) were conducted as described previously[72]. Total proteins on membranes were stained with Revert 700 (LICORbio). Antibody sources, dilutions and validation are detailed in Supplementary Data 1.

## Immunoprecipitation in plants for immunoblot detection

For anti-YFP IP experiments in Arabidopsis, freshly harvested plant tissue was homogenized in ice-cold extraction buffer (50 mM Tris–HCl, pH 8.0, 150 mM NaCl, 1 mM DTT, 1% [v/v] Triton X-100, 1x protease inhibitor cocktail [cOmplete EDTA-free, Roche]) using equal ratios of buffer to input material for all samples and a Dounce homogenizer. After centrifugation at 19,000 g and 4 °C for 5 min, the supernatant containing soluble protein was incubated with pre-equilibrated magnetic agarose anti-YFP beads (GFP-Trap, Chromo-Tek; 10 µl beads per 0.1 g plant material) for 1 h at 4 °C with rotation, washed three times with ice-cold wash buffer 1 (same as extraction buffer, but replacing Triton X-100 with 0.5% [v/v] NP-40), followed by three washes with ice-cold wash buffer 2 (50 mM Tris–HCl, pH 8.0, 150 mM NaCl), each time collecting the beads on a magnetic rack. In the last wash step, the resuspended beads were transferred to a fresh tube to remove contaminants sticking to the tube. Proteins bound to the beads after washing were eluted by boiling the beads in SDS-PAGE loading dye (50 mM Tris-HCl, pH 6.8, 2% [w/v] SDS, 100 mM DTT, 3% [v/v] glycerol, and 0.005% [w/v] bromophenol blue) at 95 °C for 5 min. Input samples are aliquots of soluble proteins not subjected to immunoprecipitation.

For the anti-HA IP in *N. tabacum*, 10-day-old seedlings were used as the starting material. Soluble protein extraction and bead incubation were carried out as described above for Arabidopsis samples but using anti-HA beads (Anti-HA high affinity, Roche). After protein binding during the incubation step, the beads were collected by centrifugation at 13,000 g for 10 s, then washed three times with

extraction buffer and three times with dilution buffer (50 mM Tris–HCl, pH 8.0, 150 mM NaCl). In the final wash step, the resuspended beads were transferred to a new tube and collected by centrifugation. Bound proteins were eluted as for Arabidopsis. If samples were RNase treated, RNase A/T1 mix (Thermo Fisher) was added to the soluble protein extracts (final concentrations: 100 µg ml$^{-1}$ for RNase A and 250 U ml$^{-1}$ for RNase T1) together with the anti-HA beads.

Transient protein expression in tobacco and subsequent IP experiments using magnetic agarose anti-YFP beads (GFP-Trap, ChromoTek) were conducted as described[82], except that wash buffer 1 (50 mM Tris–HCl, pH 8.0, 150 mM NaCl, 1 mM DTT and 0.5% [v/v] NP-40) was used for all five washes. The p19 silencing suppressor[83] was co-expressed in all cases. Each replicate derives from a separate infiltrated plant.

## Immunoprecipitation in plants followed by mass spectrometry

For IP experiments in Arabidopsis, three experiments were conducted and analyzed by MS separately. Col-0 WT plants served as controls in the three experiments. The number of biological replicates was as follows: $n = 4$ plants $P_{35S}$:*AtMKIP1-YFP*$^{+/-}$ in Col-0 background and $n = 2$ Col-0 plants for IP experiment #1; $n = 5$ plants $P_{35S}$:*AtMKIP1-YFP*$^{+/-}$ in Col-0 background and $n = 3$ Col-0 plants for IP experiment #2; $n = 5$ plants $P_{35S}$:*AtMKIP1-YFP*$^{+/-}$ in Col-0 background and $n = 3$ Col-0 plants for IP experiment #3. Total number of samples in all three experiments was 22. Some samples were re-run due to technical difficulties, but if so, only the newest run was used for downstream analysis (i.e., $n = 1$ technical replicate).

IP experiments in Arabidopsis were conducted following a procedure previously described[38] using anti-YFP µMACS magnetic beads (Miltenyi Biotec). After elution of bound proteins in 50 mM Tris-HCl, pH 6.8, and 2% (w/v) SDS, proteins were precipitated with one volume of 20% trichloroacetic acid and washed twice with cold acetone. The dry pellets were dissolved in 45 µl 10 mM Tris, 2 mM CaCl$_2$, pH 8.2 and digested by adding 5 µl of trypsin (100 ng µl$^{-1}$ in 10 mM HCl) for 30 min at 5 W and 60 °C in a microwave (Discover System, CEM). Samples were vacuum-dried and dissolved in 3% acetonitrile, 0.1% formic acid. For IP experiment #3, peptide mixtures were cleaned using the Phoenix kit (PreOmics) following the manufacturer's instructions, and the eluted samples were dried to completeness and re-solubilized in 3% acet-onitrile, 0.1% formic acid.

Samples were injected on a nanoAcquity UPLC (Waters Inc.) connected to a Q Exactive mass spectrometer (Thermo Fisher Scientific) equipped with a Digital PicoView source (New Objective). Solvent composition at the two channels was 0.1% formic acid for channel A and 0.1% formic acid, 99.9% acetonitrile for channel B. Peptides were trapped on a Symmetry C18 trap column (5 µm, 180 µm x 20 mm, Waters Inc.) and separated on a BEH300 C18 column (1.7 µm, 75 µm x 150 m, Waters Inc.) at a flow rate of 300 nl min$^{-1}$ by a gradient from 5 to 35% B in 90 min, 60% B in 5 min and 80% B in 1 min. The mass spectrometer was operated in DDA mode, acquiring full-scan MS spectra (350 − 1,500 m/z) at a resolution of 70,000 at 200 m/z after accumulation to a target value of 3,000,000, followed by higher-energy collision dissociation fragmentation on the twelve most intense signals per cycle. Spectra were acquired at a resolution of 35,000 using a normalized collision energy of 25 and a maximum injection time of 120 ms, with automatic gain control set to 50,000 ions. Singly charged and unassigned charge states were rejected. Only precursors with intensity above 8300 were selected for fragmentation. Precursor masses previously selected for MS/MS measurement were excluded from further selection for 30 s, and the exclusion window was set at 10 ppm. Data were acquired using internal lock mass calibration on m/z 371.1010 and 445.1200.

The acquired raw MS data were processed by MaxQuant[84] (v1.6.2.3), followed by protein identification using the integrated Andromeda search engine. A separate MaxQuant analysis was

performed for each of the three IP experiments. Spectra were searched against the Araport 11 database (v2017-02-01), concatenated to its reversed decoyed FASTA database and common protein contaminants, setting cysteine carbamidomethylation as fixed and methionine oxidation and N-terminal protein acetylation as variable modifications, enzyme specificity to trypsin and allowing a maximum of two missed-cleavages, and using MaxQuant Orbitrap default search settings. The maximum false discovery rate was set to 0.01 for peptides and 0.05 for proteins, label free quantification enabled and a 2-min window for matching between runs applied. Each file was kept separate in the MaxQuant experimental design to obtain individual quantitative values. Fold changes of protein abundances were computed based on intensity values reported in the proteinGroups.txt file. A set of functions implemented in the R package SRMService was used to filter for proteins with 2 or more peptides allowing for a maximum of up to 3 missing values, to normalize the data with a modified robust z-score transformation and to compute p-values using t-tests with pooled variance. If all measurements of a protein were missing in one of the conditions, a pseudo fold change was computed replacing the missing group average by the mean of 10% smallest protein intensities in that condition.

For IP experiments in tobacco followed by MS analysis, anti-HA IP using anti-HA beads (anti-HA high affinity, Roche) was conducted as for immunoblot detection (described above) without RNase treatment. Three biological replicates, each consisting of a pool of 10-day old seedlings, of the lines NtMatK C+ and NtMatK C- (as control) were assessed without technical replication. Total number of samples was 6. After the final wash, the beads were re-suspended in 45 μl digestion buffer (triethylammonium bicarbonate, pH 8.2), reduced with 5 mM tris(2-carboxyethyl)phosphine and alkylated with 15 mM chloroacetamide. Proteins were on-bead digested for 16 h at 37 °C using 5 μl of sequencing-grade trypsin (100 ng μl$^{-1}$ in 10 mM HCl, Promega). The supernatants were transferred into new tubes, the beads washed with 0.1% trifluoroacetic acid, 50% acetonitrile and combined with the first supernatant. After drying, the samples were re-solubilized in 3% (v/v) acetonitrile, 0.1% (v/v) formic acid and analyzed on the Orbitrap Exploris 480 mass spectrometer described above. Peptides were separated at a flow rate of 300 nl min$^{-1}$ by the following gradient: initial hold at 5% solvent B (0.1% formic acid in acetonitrile) / 95% solvent A (0.1% formic acid in water) for 3 min, followed by a gradient to 22% B / 65% A in 43 min, and to 32% B / 68% A in 5 min, a cleaning step at 95% B for 10 min and re-equilibration at 5% B / 95% A for 10 min. MS settings were as follows: DDA mode with a maximum cycle time of 3 s, acquisition of full-scan MS spectra (350 – 1200 m/z) at a resolution of 120,000 at 200 m/z after accumulation to a target value of 3,000,000 or for a maximum injection time of 50 ms, selection of precursors with an intensity > 5,000 for MS/MS. Ions were isolated with a 1.2 m/z isolation window, fragmented by higher-energy collisional dissociation using a normalized collision energy of 30%, spectra acquired at 30,000 resolution, with maximum injection time set to auto. Singly, unassigned and charge states higher than six were rejected. Precursor masses previously selected for MS/MS measurement were excluded from further selection for 20 s, and the exclusion window was set at 10 ppm. The data were acquired using internal lock mass calibration on m/z 371.1012 and 445.1200.

The mass spectrometry data were further processed using Philosopher[85] followed by label-free quantification based on precursor intensities using IonQuant[86]. MS2 spectra were searched against the N. tabacum UniProt reference proteome (UP000084051) concatenated to its reversed decoyed database and supplemented with MKIP1 proteins annotated in N. sylvestris [A0A1U7XZG9_NICSY, A0A1U7Y1B8_NICSY] and common contaminants by the MSFragger[76] search engine (v3.4), allowing for one missed tryptic cleavage and fixed carbamidomethylation of cysteine, variable methionine oxidation, and variable acetylation of the protein N-terminus after methionine removal.

Quantification was performed by IonQuant[86] (v1.7.17), applying the false-discovery-rate controlled match-between-runs algorithm. Fold changes and adjusted p-values were calculated using the Amica webserver[77] (v3.0.1) using the following analysis options: LFQ intensities for quantification, filtering for at least 3 MS/MS counts and at least 2 unique peptides, filtering for proteins detected in at least three replicates in at least one group, no re-normalization, imputation method "normal" (with imputation downshift = 1.8 and width = 0.3), and differential expression analysis using the implemented limma[78] framework.

## Sucrose gradient centrifugation

Approximately 60 g of leaves from 24-day-old NtMatK C+ tobacco or Col-0 wild-type Arabidopsis plants were homogenized in 500 ml homogenization buffer (50 mM HEPES-KOH, pH 7.8, 330 mM sorbitol, 2 mM EDTA, 1 mM MgCl$_2$, 50 mM sodium ascorbate, 0.25% [w/v] BSA) using a blender. The homogenate was filtered through two layers of Miracloth (Calbiochem) and centrifuged at 2000 g for 5 min at 4 °C to enrich intact chloroplasts. The pellet was resuspended in 15 ml lysis buffer (50 mM HEPES-KOH, pH 7.8, 1 mM MgCl$_2$) and incubated on ice for 1 h to lyse the chloroplasts, followed by three rounds of grinding with a glass homogenizer. The homogenate was centrifuged at 100,000 g for 20 min at 4 °C to sediment the chloroplast membranes. Proteins in the resulting supernatant were concentrated using an Amicon Ultra-4 (10 kDa molecular weight cut off) filter column by centrifugation at 4,000 g at 4 °C until the volume was reduced to 3 ml. For RNase treatment of Arabidopsis extracts, RNase A (100 μg ml$^{-1}$) and RNase T1 (250 U ml$^{-1}$) (RNase A/T1 mix; Thermo Fisher) were mixed with the stromal extracts and incubated on ice for 20 min.

Gradients of 10-30% (w/v) sucrose were prepared as previously described[65] with some modifications. Briefly, 30%, 25%, 20%, 15%, and 10% (all w/v) sucrose solutions were prepared in 30 mM HEPES pH 8.0, 10 mM MgCl$_2$, 60 mM KCl, and 5 mM DTT. Sequential 2-ml layers of each sucrose solution were carefully added on top of one another in 13.2 ml Thinwall Ultra-Clear tubes (Beckman Coulter). Each layer was frozen at −80 °C before the next one was added. The completed gradients were thawed for 16 h at 4 °C prior to use. Fresh stromal extract (500 μl, containing 2 mg protein) was carefully layered on top of the sucrose gradient. Gradients were centrifuged in an SW41 Ti rotor (Beckman Coulter) at 274,000 g (40,000 rpm) for 16 h at 4 °C, set to maximum acceleration and deceleration. Eleven 875 μl fractions were sequentially collected from the top to the bottom of the gradient. The final (12$^{th}$) fraction was pipetted up and down to resuspend any pellet.

For immunoblotting, proteins in 100-μl aliquots of each sample fraction were subjected to methanol-chloroform precipitation as described previously[87]. The precipitated proteins were dissolved in 50 μl Laemmli buffer and boiled at 95 °C for 5 min, and 10 μl was loaded for each blot. For anti-HA IP, 100 μl of each sample fraction were incubated with 5 μl anti-HA magnetic beads (Pierce) at 4 °C for 1 h with rotation, followed by three washes with chloroplast lysis buffer. The washed beads were resuspended in 20 μl Laemmli buffer, boiled at 95 °C for 5 min, and 10 μl of the eluted proteins was used for loaded for blotting.

Protein standards of the high-molecular weight gel filtration calibration kit (Cytiva) were prepared and concentrated according to the manufacturer's instructions. They were fractionated in parallel with the chloroplast extracts. Standard protein migration was assessed by Coomassie staining of methanol-chloroform precipitated fractions on SDS-PAGE gels.

## Size-exclusion chromatography and MS/MS analysis

Concentrated stromal extracts from NtMatK C+ tobacco were prepared as described for sucrose gradient centrifugation. Stromal extract (500 μl, containing 2 mg protein) was loaded onto a Superose 6 Increase 10/300 column on an ÄKTA Pure 25 system. Elution was

performed with chloroplast lysis buffer (50 mM HEPES-KOH, pH 7.8, 1 mM MgCl$_2$) at a flow rate of 0.25 ml min$^{-1}$ and 1 ml subfractions were collected. Subfractions 1–6 and 7–11 were pooled as sample fractions 1 and 2, respectively, and reduced to 1 ml each using Amicon Ultra-4 (10 kDa molecular weight cut off) columns by centrifugation at 4000 $g$ at 4 °C. Other 1-ml subfractions (numbered 3–14) were kept individually. Protein standards of the high-molecular weight gel filtration calibration kit (Cytiva) were prepared and concentrated according to the manufacturer's instructions and fractionated and eluted as the chloroplast extracts. Anti-HA IP and precipitation of 100-µl aliquots for immunoblotting were conducted as described for sucrose gradients.

For MS/MS analysis, 300 µl of selected fractions were precipitated in 80% (v/v) acetone, without technical or biological replication (i.e., $n = 1$; two samples analyzed in total). The protein pellet was resuspended in 45 µl 10 mM Tris (pH 8.0), 2 mM CaCl$_2$. Proteins were reduced with 5 mM tris(2-carboxyethyl)phosphine and alkylated with 15 mM chloroacetamide at 30 °C for 30 min, then digested using 250 ng trypsin (Sequencing Grade, Promega) for 16 h at 37 °C. After drying the peptides, they were re-solubilized in 20 µl 3% (v/v) acetonitrile, 0.1% (v/v) formic acid and spiked with iRT peptides (Biognosys). Mass spectrometry analysis was performed using the Orbitrap Exploris 480 mass spectrometer, columns and eluents A and B as described above. For each sample, 200 ng of peptides were separated at 50 °C at a flow rate of 300 nl min$^{-1}$ using the following gradient: initial hold at 5% B for 3 min, followed by a gradient from 5 to 22% B in 40 min, then 22 to 32% B in 10 min, a cleaning step at 95% B for 10 min and re-equilibration at 5% B / 95% A for 10 min (all percentages v/v). The mass spectrometer was operated in DIA mode. DIA scans covered a range from 400 to 960 m/z in windows of 8 m/z. The resolution of the DIA windows was set to 30,000, with a normalized automatic gain control target value of 1000%, the maximum injection time set to auto and a fixed normalized collision energy (NCE) of 30%. Each instrument cycle was completed by a full MS scan monitoring 396 to 1000 m/z at a resolution of 60,000.

The DIA-MS raw files were further processed using MSConvert and analyzed using DIA-NN implemented in FragPipe as described in section "Proteome analysis of β-estradiol treated plants" but searching against the *N. tabacum* UniProt reference proteome (UP000084051) concatenated to its reversed decoyed database and supplemented with MKIP1 proteins annotated in *N. sylvestris* [A0A1U7XZG9_NICSY, A0A1U7Y1B8_NICSY] and common contaminants. The two samples were analyzed together for spectral library built and protein quantification, then analyzed separately to obtain the numbers of peptides identified in the individual fractions. Proteins identified with a single peptide only were discarded from the list of quantified proteins.

## RNA co-immunoprecipitation and sequencing
Soluble cell components were extracted by homogenizing 10 g of seedlings per replicate in a pre-cooled blender using 30 ml of ice-cold extraction buffer (50 mM Tris–HCl, pH 8.0, 150 mM NaCl, 1 mM DTT, 1% (v/v) Triton X-100, 1 mM EDTA, 4 mM MgCl$_2$, 1x protease inhibitor cocktail [cOmplete EDTA-free, Roche], 40 U ml$^{-1}$ RNase inhibitor [RiboLock RNase Inhibitor, Thermo Fisher Scientific]). The homogenates were filtered through two layers of Miracloth (Milipore) and cleared by two sequential centrifugation rounds at 4000 $g$ for 10 min at 4 °C. Per replicate, 20 ml of the supernatant were incubated with 150 µl pre-equilibrated magnetic agarose anti-YFP beads (GFP-Trap, ChromoTek) for 2 h at 4 °C with rotation, washed three times with ice-cold wash buffer 1 (same as extraction buffer, but replacing Triton X-100 with 0.5% [v/v] NP-40), followed by three washes with ice-cold wash buffer 2 (50 mM Tris–HCl, pH 8.0, 150 mM NaCl). In the last wash, 10% (v/v) of the resuspended beads were collected in a separate tube and used for immunoblot analysis, while the residual beads were transferred to a new microcentrifuge tube and were processed for RNA extraction. For RNA extraction, 500 µl of input soluble protein extract

or IP-processed beads were brought to a total volume of 1 ml with DEPC-treated water, mixed with 1 ml phenol:chloroform:isoamyl alcohol (Roth) and subjected to phenol-chloroform RNA extraction as described previously[88]. Two µg input RNA was treated with TURBO DNase (Thermo Fisher Scientific) in a 50 µl reaction according to the manufacturer's instructions. From this, 600 ng purified RNA was subjected to rRNA depletion treatment as previously described[89]. The RNA was eluted in 18 µl of RNase-free water and supplemented with 2 µl RNA fragmentation buffer (400 mM Tris-acetate pH 8.3, 1 M potassium acetate, 300 mM magnesium acetate), followed by incubation at 94 °C for 5 min. Precipitate and input RNA were treated with T4 polynucleotide kinase (PNK; Thermo Fisher Scientific) as previously described[89].

Library preparation for RNA sequencing was performed using the NEXTflex small RNA-seq kit v3 (Perkin Elmer) according to the manufacturer's instructions. The resulting cDNA was amplified with a barcode incorporated in the primer and purified according to the instructions of the NEXTflex kit. Libraries were pooled for single-end 100-bp sequencing in a NovaSeq 6000 machine. Sequencing reads were adapter-trimmed and filtered for size between 18 and 60 nt using Cutadapt. The 8-nt UMI sequence in the RNA adapter was extracted, and reads were deduplicated after mapping using umi_tools. Mapping was performed using STAR with the Arabidopsis TAIR10 chloroplast genome (accession NC_000932.1) as reference, omitting the second inverted repeat and using the following parameters: outFilterMismatchNoverLmax: 0.1; alignIntronMin: 500; alignIntronMax: 1200; outSAMmultNmax: 1. Differentially abundant RNA regions and the region boundaries were identified using DiffSegR[90]. Intron domains were annotated according to the *N. tabacum* domain annotation in the CRW2 database[91].

## Total plant RNA extraction and northern blotting
Total plant RNA was extracted using GENEzol (Geneaid Biotech) according to the manufacturer instructions with minor modifications. For visual observation of rRNAs, 10 µg of total RNA was loaded onto a 1.5% agarose-formaldehyde gel (20 mM MOPS, pH 7, 8 mM sodium acetate, 1 mM EDTA, 1.5% agarose, 6% formaldehyde) and separated by size by running the gel with running buffer (20 mM MOPS, pH 7, 8 mM sodium acetate, 1 mM EDTA, 3.7% formaldehyde) at 110 V for 3 h and stained with ethidium bromide. Quantification of rRNAs was conducted using RNA ScreenTape by a TapeStation device (Agilent).

For RT-qPCR, first-strand cDNA synthesis was performed using the RevertAid First Strand cDNA Synthesis Kit (Thermo Fisher Scientific) according to the manufacturer instructions using a 1:1 mixture of random hexamer primers and oligo(dT)18 primers for cDNA synthesis. After 10-fold dilution, 1 µl cDNA was used in a 10 µl reaction of HOT FIREPol EvaGreen qPCR mix (Solis BioDyne). qPCR reactions were performed using the LightCycler 480 II system (Roche). For quantification of relative abundances of spliced and unspliced transcripts, abundances were normalized to the *AtRCE1* (*AT4G36800*) housekeeping gene[92] before normalizing to the mean of WT treated with β-estradiol. Primers are listed in Supplementary Data 1.

For northern blotting of intron/exon RNA, 10 µg of total RNA was resolved by a 1.5% agarose-formaldehyde gel and transferred to HyBond-NX Nylon membrane (GE Healthcare) by capillary transfer. The transferred RNA was then cross-linked to the membrane by two rounds of ultraviolet light irradiation (1.2 kJ each). Radiolabelled probes against target transcripts were generated using a Klenow-based labelling kit (Prime-a-Gene, Promega) using synthesized DNA primers as template in the presence of [α-$^{32}$P] dCTP. Probe hybridization, membrane washing, and membrane exposure were conducted as previously described[93], except that the hybridization and subsequent washing steps were carried out at 60 °C. The primers used to amplify the probes for analyzing intron splicing are listed in Supplementary Data 1. Northern blots for assessing rRNA processing were conducted

in the same way, except that 500 ng of total RNA was resolved and transferred to 0.45 μm neutral nylon membranes (GVS North America) by electrotransfer. Probes for analyzing plastidial rRNAs were described previously[94].

## Generation of yeast strains

Yeast strains, and constructs and primers used for creating them are detailed in Supplementary Data 1. Constructs were made following the modular cloning toolkit for yeast[95]. For *AtMKIP1*, the CDS of splice form 2 was used in all experiments. Plasmids were validated as described for plant plasmids. Yeast transformation, transformant selection via hygromycin resistance or uracil prototrophy, strain purification and confirmation of correct genomic integration by PCR were conducted as previously described[34].

For expression of YFP-tagged orthologous BEs, the CDS of BEs (without their cTPs, unless cTP prediction was weak; Supplementary Data 1) were cloned as part 3a into pYTK001 entry vector[95] via *Bsm*BI. They were then individually assembled with the other parts (the strong constitutive *CPW2* promoter, a C-terminal *eYFP* tag, and the *CYC1* terminator) via *Bsa*I into the pre-assembled yeast integrative vector pBP296 (ref. 34), which targets the intergenic *XII-5* yeast locus[96]. The resulting vectors containing single transcription unit (TU) for BE expression were transformed into the haploid CEN.PK113-11C yeast strain "578"[34] (*MATa MAL2-8^C SUC2 his3Δ malx2 glc3Δ gsy2Δ glg1Δ glg2Δ gph1Δ gsy1::pGAL1-EcglgC-TM-HA-tCYC1 bar1Δ XII-2::pCWP2-mCherry-tUPT7:KanR gdb1::pUBI4-AtSS1-tRPL3*).

For IP experiments in yeast, final yeast integrative vectors were targeted to the intergenic yeast loci[96] *XII-5* (for testing homo-oligomerization of *At*MatK) or *XII-2* (all other tests)[96]. We therefore used either single TU constructs (for control strains expressing only a single test protein) or multi TU constructs (for pair-wise interaction tests; Supplementary Data 1). The *AtMatK* CDS (codon optimized for Arabidopsis and containing the edit frequently observed in *AtMatK* RNA[97], resulting in H236Y conversion) was first cloned as part 3a into pYTK001 via *Bsm*BI (pBP643), then assembled with parts for the promoter, a C-terminal tag and a terminator via *Bsa*I into the single-TU yeast vectors pBP662 (for the $P_{GAL1}$:*AtMatK-3xHA*:$T_{CYC1}$) or pBP773 (for $P_{GAL1}$:*AtMatK-FLAG*:$T_{CYC1}$). For expression of *At*MKIP1-YFP and *At*BE3-YFP without their cTPs, the same TUs as for orthologous BE expression ($P_{CWP2}$:*AtMKIP1-YFP*:$T_{CYC1}$ from pBP595; $P_{CWP2}$:*AtBE3-YFP*:$T_{CYC1}$ from pBP596) were used but targeting them to locus *XII-2*. An additional single TU vector pBP774 was produced in an equivalent manner for expressing on $P_{CWP2}$:*AtMKIP1-3xHA*:$T_{CYC1}$. For expression of *At*MKIP1 fragments, the corresponding regions (AA 120-333 for CBM-MSR; AA 178-333 for MSR; numbers based on the full-length *At*MKIP1 protein including its cTP) were cloned as part 3a into pYTK001, then assembled into single-TU vectors as the native *At*MKIP1 protein. The ΔMSR-version of *At*MKIP1 was created by amplifying the *At*MKIP1(ΔMSR) fragment from the vector AtMKIP1(ΔMSR)_pUBC-YFP and assembling it into *Hind*III pre-digested pBP595 using the NEBuilder HiFi DNA Assembly kit (New England Biolabs). N-terminal (AA 1-381) and C-terminal (AA 357-526) *At*MatK segments were created by deleting the other MatK regions in pBP662 by DNA assembly as above.

For pair-wise interaction tests, multi-TU integrative vectors were fused via *Bsm*BI using the preassembled integrative vectors pBP492 or pBP249, single-TU vectors and primer-based spacers occupying empty positions. Final single- and multi-TU vectors were *Not*I digested and transformed into the haploid CEN.PK113-11C strains "344"[34] (*MATa MAL2-8C SUC2 his3Δ glc3Δ gsy2Δ glg1Δ glg2Δ*), or close descendants (Supplementary Data 1).

## Light microscopy and branching enzyme activity gels in yeast

For light microscopy of iodine stained yeast cells, yeasts were inoculated in complex medium with 2% (w/v) glucose (YPD) until saturation, diluted in complex medium with 2% (w/v) galactose (YPGal, to induce

expression of $P_{GAL1}$-driven ADPglucose pyrophosphorylase) to an OD of ~0.3 in Erlenmeyer flasks. After shaking for 5.75 h at 30 °C and 260 rpm, cells were harvested by centrifugation, washed with water, flash-frozen in liquid $N_2$ and stored at -80 °C. Cells were thawn on ice, stained by mixing with Lugol's solution in a 1:1 ratio, and immediately imaged using an Axio Imager.Z2 light microscope equipped with a 100X oil-immersion objective (Zeiss).

For protein analysis, yeasts were grown in the same way, but the main cultures were harvested already after 3 h shaking in YPGal. Total and soluble protein extracts for immunoblotting were prepared as previously described[34]. Immunoblotting was conducted as described above for plant proteins. Native protein extraction and native PAGE monitoring BE activities (zymograms) by phosphorylase *a* activity stimulation were conducted as previously described[25], with the relative load of 1.0 referring to 15 μg soluble proteins and running the gels with 100 V for 3.5 h at 4 °C.

## Immunoprecipitation experiments in yeast

Yeasts were inoculated in YDP, grown for ~6 h at 30 °C and 260 rpm, then inoculated to an OD of ~0.015 in Erlenmeyer flasks and shaken for 16 h at 30 °C and 260 rpm. Cells were harvested by centrifugation and once washed in water. Fresh cell pellets were resuspended in 3.3 volumes of native protein extraction buffer (0.1 M MOPS, pH 7.75, 1% [v/v] Triton X-100, 1 mM EDTA, 4 mM $MgCl_2$, protease inhibitor [Complete EDTA-free; Roche]) and 4.3 volumes of glass beads (acid-washed, 425–600 mm diameter) and homogenized by vortexing for 30 min at 4 °C. The homogenate was transferred without glass beads to a fresh tube and centrifuged at 16,000 g for 5 min at 4 °C. Equal amounts of soluble proteins present in the supernatant were mixed with pre-blocked and/or washed magnetic beads. In case of anti-HA beads (Anti-HA Magnetic Beads, Pierce), beads had been pre-blocked by incubation for 30 min at 20 °C in TBS-T (Tris-buffered saline, 0.1% [v/v] Tween-20) supplemented with 5% (w/v) bovine serum albumin (BSA), then washed with TBS-T and resuspended in native protein extraction buffer. Magnetic agarose anti-YFP beads (GFP-Trap, ChromoTek) had been prepared by washing them once in wash buffer (50 mM Tris-HCl, pH 8, 0.15 M NaCl, 1 mM DTT, 1% [v/v] Triton X-100) and resuspending them in native protein extraction buffer. The protein-bead mixture was incubated on a rotating wheel at 4 °C for 1 h, then washed six times with wash buffer (for anti-YFP beads) or TBS-T (for anti-HA beads), using a magnetic stand for bead recovery. In the last wash step, beads were transferred to a fresh tube to remove proteins adhering to the plastic walls. Proteins bound to the beads were eluted by boiling in SDS-PAGE loading buffer as described for Arabidopsis IP above. Input samples are aliquots of soluble proteins not subjected to immunoprecipitation. Replicates derive from individual main cultures.

## Sequence analyses and protein structure prediction

Starch-branching enzyme protein sequences were identified by BLASTp using *At*MKIP1, *At*BE2, *At*BE3, and rice BEI (a class I BE) as query in multiple genome and transcriptome databases. Sequences were manually curated to remove duplicates and sequences shorter than 200 amino acids very likely constituting fragments. Sequences were aligned by MAFFT (v7)[98] with the E-INS-I algorithm. Phylogenetic analyses were conducted in PhyloBayes MPI (v1.8c) using the "CAT + GTR" substitution model. Two independent runs were performed, each cycling until the sampled tree had stabilized and reached satisfactory convergence (maxdiff <0.3). A convergent tree was calculated from the two runs and the posterior probabilities computed, discarding the first 500 cycles ( < 5% of total cycles) of each run as burn-in. The unrooted phylogenetic tree was plotted using Interactive Tree of Life[99] (v6).

For protein conservation analysis, protein sequences of each BE class (defined by the phylogenetic tree) were aligned using MAFFT (v7)[98] with the E-INS-I algorithm. Motif logos of regions of interest were

constructed using WebLogo 3[100]. BE domains were predicted by SMART[101], Interpro[102] and a locally installed version of ChloroP[103]. Disordered regions were predicted by AIUPred[104]. The MSR was inferred from the multiple sequence alignment. The conserved regions in the central catalytic domain were defined as previously[33]. RT motifs and the X domain of *At*MatK base on the alignment to the *Nt*MatK domains annotated previously[17], except that the X domain was slightly elongated (AA 382-495 instead of 382-483) to fully include an α-helix predicted by AlphaFold3. Protein structures without their predicted cTP (Supplementary Data 1) were predicted by AlphaFold2 (for monomeric MKIP1;[105]) or AlphaFold3 (for protein complexes;[106]). Interfaces in the four-subunit complex were detected by AlphaBridge[107]. Molecular graphics and analyses were performed with UCSF ChimeraX (v1.10)[108]. The predicted binding sites are contacts between *At*MatK and *At*MKIP1 within 4 Å distance with a maximum PAE value of 5 Å. Only binding sites containing at least 3 contacts are shown, with a maximum gap of 2 AA between contacts allowed.

## Accession numbers

The Arabidopsis Genome Initiative gene codes associated with this study are: AT3G20440 (*AtMKIP1/AtBE1*), ATCG00040 (*AtMatK*), AT1G07320 (*AtuL4c*), AT5G16715 (*AtValRS2*), AT3G14900 (*AtEMB3120*), AT2G36390 (*AtBE3*). Accession numbers for BE orthologs expressed in yeast are given in Supplementary Data 1.

## Statistical analyses

The data shown in Figs. 8f, 9c-e and Supplementary Fig. 15 were first evaluated for normal distribution using the Shapiro-Wilk Test and tested for equal variance using Bartlett's test. Since the data did not meet the assumptions of normal distribution and equal variance, the non-parametric Kruskal-Wallis test was used for determining statistically significant differences between the groups, followed by a two-sided Dunn's post hoc multiple comparisons test and Bonferroni *p*-value adjustment to identify the specific groups that are significantly different. Statistical analyses of the mass spectrometry and RNA sequencing data is described in the corresponding sections and figure legends.

## Reporting summary

Further information on research design is available in the Nature Portfolio Reporting Summary linked to this article.

## Data availability

All data needed to evaluate the conclusions in this paper are present in the paper and/or its supplementary materials. Source Data are provided with this paper. The proteomics data are freely available at the ProteomeXchange Consortium via the PRIDE[109] partner repository with the identifiers PXD060108 (https://www.ebi.ac.uk/pride/archive/projects/PXD060108) (IP data of *At*MKIP1-YFP), PXD060055 (https://www.ebi.ac.uk/pride/archive/projects/PXD060055) (IP data of *Nt*MatK-HA), PXD069946 (https://www.ebi.ac.uk/pride/archive/projects/PXD069946) (MS/MS analysis of SEC fractions), and PXD067382 (https://www.ebi.ac.uk/pride/archive/projects/PXD067382) (Proteomics after silencing of *AtMKIP1* or *AtuL4c*). The RNA sequencing data of RNAs bound to *At*MKIP1-YFP are publicly available at NCBI GEO[110] with the accession number GSE284378. Microscopy images are freely available at the ETH Research Collection (https://doi.org/10.3929/ethz-c-000795755). New genetic material (transgenic Arabidopsis and yeast lines) and vectors will be made available to the scientific community upon request. Source data are provided with this paper.

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

## Acknowledgements

We thank Martha Stadler for excellent technical support, Andrea Ruckle for help with plant cultivation, Christian Schmitz-Linneweber (HU Berlin) for providing us with the *Nt*MatK C+ and C- tobacco lines and Ari Pekka Mähönen (University of Helsinki) for providing us with the pH7m34GW vector. We thank Bernd Roschitzki, Sibylle Pfammatter, Paolo Nanni and Tobias Kockmann from the Functional Genomics Center Zurich (FGCZ) for help with the proteomics analyses. AlphaFold2 structure predictions were performed on the ETH Zurich Euler computing cluster. RT-qPCR and TapeStation data were produced in collaboration with the Genetic Diversity Centre (GDC), ETH Zurich. This project has received funding from the European Union's Horizon 2020 research and innovation programme under the Marie Sklodowska-Curie (MSC) grant agreement No 847585 (to S.C.Z. and B.P.) and from the Vontobel Foundation (to B.P.). R.Z. is supported by the Max Planck Society and the DFG grant ZO 302/5-1.

## Author contributions

B.P., Y.L., and S.C.Z. designed the research. Y.L. conducted most of the research. Y.G., A.F., M.A., A.G., M.G., C.L., M.S., and B.P. collected and analyzed data. Y.L. prepared the figures. B.P. and Y.L. wrote the manuscript. S.C.Z, A.G., R.Z., M.S., and Y.G. edited the manuscript. B.P., S.C.Z., and R.Z. supervised the research and acquired funding. All authors read and approved the manuscript.

## Funding

## Competing interests

The authors declare no competing interests.
