## [Transparent Peer Review file · Nature Communications]

Maturase K forms a plastidial splicing complex with a neofunctionalized branching enzyme

Corresponding Author: Dr Barbara Pfister

Version 0:

Reviewer comments:

Reviewer #1

(Remarks to the Author)

In this manuscript, Liang et al. describe the identification and molecular characterization of MKIP1 a chloroplast protein interacting with the plastid groupII intron splicing factor MatK. Unexpectedly, MKIP1 shares homologies with unrelated starch branching enzymes (BEs). However, the authors unambiguously demonstrate that MKIP1 has lost its BE activity and, instead, has acquired a new function in chloroplast RNA metabolism. Moreover, the comprehensive genetic and molecular work clearly shows that MKIP1 associates with intron RNAs and is involved in their splicing. The manuscript reads well and the findings provide new insights into the evolution and molecular players in organellar splicing. However, before publishing a few points might be considered.

Whereas the genetic data convincingly reveals a role of MKIP1 during splicing, its precise function remains a bit diffuse. This is mainly due to the lack of experimental data on the detected MatK/MKIP1 RNP complex. What is the molecular size of this complex (e.g. in native PAGE)? What is the stoichiometry of the subunits? For instance, the blue coloring in the lower-right corner of the predicted aligned error (PAE) plot (Fig. 4a), corresponding to intermolecular regions of AtMKIP1, suggests potential homodimerization or higher-order oligomeric states for AtMKIP1. To investigate such alternative conformations, the application of AlphaFold3 would be helpful (Abramson et al., 2024). Additionally, AlphaFold3 could be utilized to predict protein–nucleic acid interactions, enabling a more detailed understanding of the recruitment order within this functional complex. Such predictions could significantly refine the proposed model illustrated in Fig. 8.

In line with this, does recombinant MKIP1 produced in yeast has RNA binding affinity on its own?

Minor points:

- Fig.1b: It is stated in the text (line 350) that AtMKIP1-YFP lacked a punctuate sub-organellar localization pattern typical for nucleoids. However, to this reviewer some chloroplasts in Fig. 1b do display a punctuate pattern. Maybe imaging can be improved.

- Fig. 1e: In the Figure legend, it should be given what the difference between the two last images from true leaves is.

Reviewer #2

(Remarks to the Author)

review to Nat. Comm (NCOMMS-24-81917) "Maturase K forms a plastidial splicing complex with a neofunctionalized branching enzyme"

Plastids typically harbor introns in various essential genes that must be removed post-transcriptionally to allow the functionality of the mature transcripts. The work of Yuanyuan Liang et al focuses on MKIP1, an essential gene encoding a plastid-localized homolog of starch-branching enzymes (BEs). The authors indicate that MKIP1 lacks BE activity and instead acquired features that enable it to facilitate gII's processing in coordination with the intron-encoded MatK proteins. The two proteins can interact with one another independently of gII intron RNAs. The authors further indicate a role for MKIP1's in RNA-binding and established embryo rescue or (estradiol-inducible) micro-RNA knockdown lines to investigate its roles in cp-RNA metabolism. The RNA profiles showed that MKIP1 mutant lines are indeed affected in gIIA's splicing (excluding ClpP), which is similar to suggested roles of MatK protein (Zoschke et al 2020). Additionally, co-IPs indicated that AtMKIP1 is associated with multiple group IIA introns in vivo, again in a similar manner to those of MatK. Overall, this reviewer finds that the data is original and strengthens the study's impact on RNA metabolism in the organelles of plants. The findings of

MKIP1 roles in plastid gII intron splicing and interaction with MatK are novel and convincing, although I find the suggested model of action less supported by the data and previous in vitro analyses of MatK. Likewise, the work, figures and the data organization are definitely of good quality. The clarity of the methodology and the overall quality of the data obviously make it a valuable contribution to the field. I have some comments to the authors that may improve the data and reading of the MS.

Specific comments regarding the MS data and text:

1. The analysis of the mutants needs further support. i.e., as indicated Fig 1 relates to "hemi-complemented" (genetically-rescued) ABI3 lines, while the RNA and protein profiles relate to estradiol-inducible amiR lines. I find this confusing. more importantly, it would be more convincing to show that all the data, e.g., organellar ultrastructural morphologies, as well as the plastidial RNA and protein profiles are done for all the genetic lines (amiR and ABI3 lines)
2. The effects of the mutations of RNA metabolism are convincing (excluding comment #1), but less so for the overall protein profiles. The authors indicate translation defects for a few plastid-encoded proteins. One would assume that altered processing of transcripts that correspond to tRNAs and ribosomal factors (Fig. 7 and Fig. S10) would generally affect organellar gene expression. Unclear whether this is indeed supported by the mutant's protein data.
3. The lower blot in Fig 7 is cut for the corresponding RbcL bands! it would also be more convincing to show the complete blots.
4. Also, regarding comments above - how do the relatively mild effects seen in the RNA and protein data correlate with the strong defects in plant physiology?
5. The background on group II introns is comprehensive but assumes prior knowledge. Adding a short/brief explanation of their function and relevance in plants could help readers that are unfamiliar with the topic.
6. The transition to discussing MatK and its divergence from bacterial maturases is a bit abrupt. Consider smoother transitions.
7. MKIP1 gene locus, and gene expression. Notably, the AT3G20440 locus at TAIR database has alternative splicing variants, one of which affects the N-terminal region (AT3G20440.3). Is there a reason that the authors focus on one of these for the genetic analysis AT3G20440.2?
8. How the authors explain the association of MKIP1 with itself and with the other splicing variants (IP MS data in suppl. data)? do they all correspond to the same peptides? or rather co-IPed the two paralogous variants.
9. Fig. 1 - Localization analysis. The confocal microscopy requires additional controls (mito's , nuclear at cetera). Note that MKIP1 was identified in a nuclear MS/MS analysis (although may correspond to chloroplast contamination).
10. Following their data, and in light of Fig 6c (MS results lines 292-294), the authors need to show the global effects of MKIP1 mutations on ribosome biogenesis effects (e.g., sucrose gradients, RNase protection assays of plastidial rRNAs) as well as on chloroplast translation (i.e., MS/MS).
11. The model in Fig 8 is not supported, and might be wrong. Other than a few canonical MAT's, the association of MatK with group IIA intron RNA targets is indeed currently unknown. However, MatK was previously shown to demonstrate some specificity in its association with its defined gII's targets in vitro, this in the absent of additional protein cofactors (see Liere, K. and Link, G. (1995) RNA-binding activity of the Matk protein encoded by the chloroplast trnK intron from mustard (Sinapis-Alba L). *Nucleic Acids Res*, 23, 917-921.; Barthet, M.M., Pierpont, C.L. and Tavernier, E.K. (2020) Unraveling the role of the enigmatic MatK maturase in chloroplast group IIA intron excision. *Plant Direct*, 4, e00208. MatK protein contains part of the RT0 sequence motif (a region with importance for specific gII RNA binding), although most of this region (and other T elements) seem indeed missing from MatK.

Some other minor remarks:

1. The abstract is well-written, but some sentences are lengthy and could benefit from simplification for clarity.
2. The authors may want to briefly explain at the first paragraph in the intro/abstract how At-MKIP1 was originally identified as a MatK interacting partner (co-IPs?)
3. "Arabidopsis MKIP1 specifically co-precipitates all known intron targets of MatK." Could be rephrased (e.g., "In Arabidopsis, MKIP1 co-precipitates with all known intron targets of MatK.")
4. The concluding statement about the N-terminus of MatK could be clearer if rephrased.
5. The introduction is quite clear but occasionally verbose. For example: "Chloroplast gene expression involves the splicing of ~20 introns, a process essential for plant viability." Consider to alter and to combine with the subsequent sentence for a smoother transition.
6. The section on MatK's conserved X domain could use reorganization for better logical flow.
7. Results line 177 - EMB3120 is annotated at TAIR as - SNOW-WHITE LEAF1, SWL1 (see REF of Yue Wang et al Crucial role of SWL1 in chloroplast biogenesis and development in Arabidopsis thaliana. *Plant Cell Rep* 43, 135 (2024). <https://doi.org/10.1007/s00299-024-03210-1>).

Version 1:

Reviewer comments:

Reviewer #1

(Remarks to the Author)

In the new version of the manuscript, the authors appropriately addressed all of my points.

Reviewer #2

(Remarks to the Author)
review to NCOMMS-24-81917A:

Overall assessment

In the revised MS, I find that the authors have responded constructively to the comments. Some experimental limitations remain obvious (e.g., due to seedling lethality), but these are now more clearly stated, and additional data and analyses strengthen the original work. Major conceptual concerns, including the unsupported model, have been adequately addressed. Overall, the revised manuscript is improved and the responses are aligned with the reviewer's comments. Below I attached a point-by-point comments, highlighting where responses are fully aligned, adequately justified despite limitations, or still somewhat constrained but still otherwise acceptable.

- **Mixing genetic systems (ABI3 vs amiR lines):** Authors acknowledge the confusion, explain biological/technical constraints (seedling lethality, tissue limitation), and justify why deep molecular analyses focus on amiR lines compensate by adding TEM for amiR lines, expanding proteomics and rRNA analyses, explicitly stating limitations in the Results. Although this part may not fully meet the ideal request (same assays in all lines), the response is scientifically justified which is acceptable in my opinion.
- **Protein-level effects vs RNA defects:** Authors provide in their revised MS a quantitative MS/MS proteomics, showing to reduced plastid ribosomal proteins, reduced plastome-encoded proteins, and dosage-dependent effects across amiR lines. This more convincingly addresses the main concern and further strengthens the manuscript.
- **RbcL blot (Fig. 7):** I find that the revised MS and authors response adequately addressed the comments. Authors give a biologically coherent explanation, with the timing of induction, differential sensitivity of developing vs mature tissues, and a stronger phenotype under early induction. The authors further avoid overinterpretation and acknowledge pleiotropy. This is quite convincing and seems consistent with what is known and established in plastid biology.
- **Group II intron background:** Authors expanded the introduction as requested.
- **Abrupt transition to MatK discussion:** Minor but important stylistic fix implemented as advised.
- **MKIP1 splice variants and locus choice:** The revised MS and author response are well aligned with the comments. RNA-seq support proteomics evidence, functional complementation, and a clear justification for focusing on AT3G20440.2. I find that this adequately resolves the concern.
- **MKIP1 self-association / paralog ambiguity:** Authors clarify peptide mapping ambiguity. They provide additional experimental evidence (tobacco transient IPs) ruling out self-oligomerization. As in the case of the previous comment, this directly answers my comment.
- **Localization controls:** Authors add MitoTracker, DAPI, improved confocal imaging. So they fully address the prior concern. This strengthens the localization claim.
- **Global ribosome biogenesis and translation effects:** In my comments I suggested broader evidence (e.g., rRNAs, translation). Authors add here a comprehensive rRNA northern blots, some proteomics data, comparative logic with known ribosome mutants, they explicitly frame these as secondary effects, which is scientifically appropriate. While sucrose gradients might have been ideal, the combination of data provided herein in the revised MS reasonably "satisfies" the concern.
- **Unsupported model (Fig. 8):** Authors fully accept the criticism (of both rev's) and remove the model. They reason MatK binding with prior literature and clearly separate speculation from data.
- **Minor remarks section:** regarding Abstract clarity, Intro verbosity, Terminology, Citation additions, and Rephrasing requests - these have been adequately addressed and appropriately revised.

Responses to the reviewers

We would like to thank both reviewers for their valuable suggestions, which we believe have helped us improve the manuscript. We were able to address almost all concerns except in a few cases where we encountered technical limitations. Additionally, we updated all multimer predictions from AlphaFold2 to AlphaFold3 to ensure consistency.

Reviewer #1 (Remarks to the Author):

In this manuscript, Liang et al. describe the identification and molecular characterization of MKIP1 a chloroplast protein interacting with the plastid groupII intron splicing factor MatK. Unexpectedly, MKIP1 shares homologies with unrelated starch branching enzymes (BEs). However, the authors unambiguously demonstrate that MKIP1 has lost its BE activity and, instead, has acquired a new function in chloroplast RNA metabolism. Moreover, the comprehensive genetic and molecular work clearly shows that MKIP1 associates with intron RNAs and is involved in their splicing. The manuscript reads well and the findings provide new insights into the evolution and molecular players in organellar splicing. However, before publishing a few points might be considered.

Whereas the genetic data convincingly reveals a role of MKIP1 during splicing, its precise function remains a bit diffuse. This is mainly due to the lack of experimental data on the detected MatK/MKIP1 RNP complex. What is the molecular size of this complex (e.g. in native PAGE)? What is the stoichiometry of the subunits? For instance, the blue coloring in the lower-right corner of the predicted aligned error (PAE) plot (Fig. 4a), corresponding to intermolecular regions of AtMKIP1, suggests potential homodimerization or higher-order oligomeric states for AtMKIP1. To investigate such alternative conformations, the application of AlphaFold3 would be helpful (Abramson et al., 2024).

Response: *We thank the reviewer for the excellent recommendation to characterize the molecular weight and composition of the MatK complex further and to implement more AlphaFold3 predictions. While we were unable to detect the complex subunits in native PAGE, we could analyse the migration of the complex in sucrose density gradients and its elution using size exclusion chromatography. For this, we used the tobacco NtMatK C+ line, which allowed us to detect MatK-HA by western blotting, as well as Arabidopsis. These data show that NtMatK, NtMKIP1, NtValRS2 and NtEMB3120 co-migrate into high-molecular-weight fractions, suggesting that they together form a stable complex (**new Fig. 4** and **new Supplementary Fig. 9**). This is consistent with AlphaFold3 prediction of a four-subunit complex with a 1:1:1:1 stoichiometry of AtMatK, AtMKIP1, AtValRS2 and AtEMB3120 with relatively high confidence ($ipTM = 0.7$; **new Fig. 5**).*

*We refrain from estimating an exact molecular weight of the complex from these methods since these estimations can be biased for non-globular protein complexes. However, we tested for possible self-association of AtMKIP1, AtMatK, AtValRS2 and AtEMB3120 after expression in tobacco or in yeast (**new Supplementary Fig. 10**). None of the proteins showed significant self-association. This is again consistent with the AlphaFold3 predictions, which did not support homo-oligomerization for any subunit. AlphaFold3 predictions also did not support the association of a second molecule of either subunit in the presence of the other complex components (**new Supplementary Fig. 10**). Collectively, these data suggest that the complex exists in the form of a tetramer with one subunit of each MatK, MKIP1, ValRS2 and EMB3120. The observed molecular weight ranges of the complex by size-*

exclusion chromatography and sucrose density gradient centrifugation would be certainly plausible for a tetramer (~340 kDa, not counting any associated RNA fragments). While more details on the composition and structure would be very interesting, we feel that they are better suited for follow-up work.

We would like to clarify that the PAE plot of the AtMatK-AtMKIP1 heterodimer (now in Fig. 6d) only indicates the confidence in the relative placement of the AtMatK and AtMKIP1 residues, and does not allow any conclusions to be drawn about the potential homo-oligomerisation of either subunit.

AlphaFold3 could be utilized to predict protein-nucleic acid interactions, enabling a more detailed understanding of the recruitment order within this functional complex. Such predictions could significantly refine the proposed model illustrated in Fig. 8.

Response: Although AlphaFold3 is better than previous versions at predicting RNA folding, these predictions remain difficult, particularly for large RNAs such as group II introns. Unfortunately, the plastidial group II intron structures predicted by AlphaFold3 did not resemble the experimentally determined structures of related bacterial group II introns. AlphaFold3 was also unable to predict smaller intron fragments of around 50 nucleotides with reasonable confidence. Given the high likelihood that RNA structures are not being predicted correctly, we lean away from predicting their interaction with the MatK complex and feel that future work should aim to experimentally determine the structure of the RNP complex. In this light, we agree that the model originally presented in Fig. 8 is not sufficiently supported and have thus removed it. We have also added a statement in the discussion regarding the difficulties in modelling group IIA introns and their association with the complex.

In line with this, does recombinant MKIP1 produced in yeast has RNA binding affinity on its own?

Response: We thank the reviewer for this useful suggestion. We spent a considerable amount of time obtaining soluble proteins of good purity and testing different conditions for RNA binding. Ultimately, we managed to express and purify AtMKIP1 (and AtBE3 as a control) fused to an N-terminal 6x-His-SUMO tag in *E. coli*, albeit yielding relatively low concentrations. We tested each domain of the Arabidopsis *trnK* intron for RNA binding using electrophoretic mobility shift assays. We repeatedly observed the association of the tagged AtMKIP1 with the complete DVI, but not with any of the other tested RNA regions. No significant binding of AtBE3 to DVI was observed. However, as the extent of AtMKIP1 binding varied between experiments and binding saturation could not be observed, we could not estimate affinities. We believe that establishing AtMKIP1's RNA binding would require further optimization and the combination of multiple techniques (ideally incorporating structural information of the RNP complex – see above) to achieve conclusive, robust, and biologically meaningful results. This follow-up work is certainly very interesting but is beyond the scope of this manuscript.

Minor points:

- Fig. 1b: It is stated in the text (line 350) that AtMKIP1-YFP lacked a punctuate sub-organellar localization pattern typical for nucleoids. However, to this reviewer some chloroplasts in Fig. 1b do display a punctuate pattern. Maybe imaging can be improved.

Response: We agree that AtMKIP1-YFP appeared somewhat inhomogeneously distributed within the chloroplasts. This may reflect an uneven distribution of AtMKIP1-YFP within the stroma due to the presence of thylakoid membranes, which differs from the distinct punctae

typically observed for nucleoids (see, for example, Fig. 1A from Powikrowska et al., 2014. *Dynamic composition, shaping and organisation of plastid nucleoids. Frontiers in Plant Science*, <https://doi.org/10.3389/fpls.2014.00424>). To clarify, we have replaced this image with more detailed micrographs. In response to a suggestion from Reviewer #2, these also show the locations of the mitochondria and nuclei (Fig. 1b and new Supplementary Fig. 3). The new data indicate that, under the tested conditions, AtMKIP1 localises exclusively to the plastids, where it appears to be distributed fairly evenly within the stroma.

- Fig. 1e: In the Figure legend, it should be given what the difference between the two last images from true leaves is.

Response: We have now indicated some of the structures observed within the plastids with arrows and added the following statement to the legend: “In the true leaves from rescued *mkip1-1^{-/-}* plants, plastids typically contained collapsed thylakoids (red arrow) and vesicular structures (yellow arrow). Occasionally, larger thylakoid stacks were observed (outer right image).”

Reviewer #2 (Remarks to the Author):

Plastids typically harbor introns in various essential genes that must be removed post-transcriptionally to allow the functionality of the mature transcripts. The work of Yuanyuan Liang et al focuses on MKIP1, an essential gene encoding a plastid-localized homolog of starch-branching enzymes (BEs). The authors indicate that MKIP1 lacks BE activity and instead acquired features that enable it to facilitate gII' s processing in coordination with the intron-encoded MatK proteins. The two proteins can interact with one another independently of gII intron RNAs. The authors further indicate a role for MKIP1' s in RNA-binding and established embryo rescue or (estradiol-inducible) micro-RNA knockdown lines to investigate its roles in cp-RNA metabolism. The RNA profiles showed that MKIP1 mutant lines are indeed affected in gIIA' s splicing (excluding ClpP), which is similar to suggested roles of MatK protein (Zoschke et al 2020). Additionally, co-IPs indicated that AtMKIP1 is associated with multiple group IIA introns in vivo, again in a similar manner to those of MatK. Overall, this reviewer finds that the data is original and strengthens the study's impact on RNA metabolism in the organelles of plants. The findings of MKIP1 roles in plastid gII intron splicing and interaction with MatK are novel and convincing, although I find the suggested model of action less supported by the data and previous in vitro analyses of MatK. Likewise, the work, figures and the data organization are definitely of good quality. The clarity of the methodology and the overall quality of the data obviously make it a valuable contribution to the field. I have some comments to the authors that may improve the data and reading of the MS.

Specific comments regarding the MS data and text:

1. The analysis of the mutants needs further support. i.e., as indicated Fig 1 relates to “hemi-complemented” (genetically-rescued) ABI3 lines, while the RNA and protein profiles relate to estradiol-inducible amiR lines. I find this confusing. more importantly, it would be more convincing to show that all the data, e.g., organellar ultrastructural morphologies, as well as the plastidial RNA and protein profiles are done for all the genetic lines (amiR and ABI3 lines)

Response: We thank the reviewer for these useful suggestions. Unfortunately, there is very little tissue available from the seedling-lethal $P_{ABI3}::AtMKIP1-mCitrine^{+/+}/mkip1^{-/-}$ plants. Despite pooling homozygous mutant seedlings from multiple plates, we were unable to recover sufficient RNA to test for splicing deficiencies using northern blotting with multiple replicates. Nevertheless, we analysed all non-tRNA introns using RT-qPCR and identified significant splicing defects in the MatK/MKIP1-associated group IIA introns (i.e. in intron 2 of *rps12* and the introns in *rpl2* and *atpF*). These splicing defects were approximately twice as strong as those observed in the *amiR-mkip1* lines. These data are consistent with the strong visual phenotypes and with a role for AtMKIP1 in the splicing of these introns. However, given the seedling lethality of this line, we expect this tissue to exhibit pleiotropic defects, including severe impairment of plastidial translation. This makes identification of the primary molecular defect difficult, particularly since we lack an appropriate control line with a primary defect in plastidial translation. For these reasons, we did not include the limited splicing data we have in the manuscript.

None of the problems described above apply to the *amiR-mkip1* lines: material is sufficient, so no pooling is required, and the inducible system allowed us to harvest the tissue before strong pleiotropic effects develop. In addition, we have an appropriate control line (*amiR-uL4c*) available to account for the mild defects in plastidial translation. In the revised version of the manuscript, we have included transmission electron micrographs of *amiR-mkip1* from an earlier, comparable batch, which show that the plastidial ultrastructure is altered and contains fewer thylakoid stacks (**new Supplementary Fig. 12**). In response to the reviewer's other concerns, we have expanded the molecular analyses to include a thorough assessment of proteome changes (**Fig. 8** and **new Supplementary Fig. 13**), as well as rRNA analyses (**new Supplementary Fig. 14**). We have also added a statement at the beginning of the silencing results section explaining the limitations of the $P_{ABI3}::AtMKIP1-mCitrine^{+/+}/mkip1^{-/-}$ lines and the linkage between plastidial splicing and translation.

2. The effects of the mutations of RNA metabolism are convincing (excluding comment #1), but less so for the overall protein profiles. The authors indicate translation defects for a few plastid-encoded proteins. One would assume that altered processing of transcripts that correspond to tRNAs and ribosomal factors (Fig. 7 and Fig. S10) would generally affect organellar gene expression. Unclear whether this is indeed supported by the mutant's protein data.

Response: To analyse the impact of MKIP1 silencing on plastidial gene expression and abundance of plastome-encoded proteins at larger scale, we conducted total proteome analysis by MS/MS in the three *amiR* lines and WT control. Indeed, most of the quantified plastidial ribosomal proteins of the large and small ribosomal subunits (**new Supplementary Fig. 13c**) are significantly reduced in *amiR-mkip1-1*. The same applies to the quantified plastome-encoded proteins (**Fig. 8b**). Reduced levels of ribosomal and plastome-encoded proteins are also apparent in *amiR-mkip1-2* but milder and mostly not statistically significant, consistent with the generally milder reduction in MKIP1 and phenotypes of this line. These data indicate that silencing of MKIP1 indeed has widespread negative effects on plastidial gene expression.

3. The lower blot in Fig 7 is cut for the corresponding RbcL bands! it would also be more convincing to show the complete blots.

Response: Unfortunately, the membrane had been physically cut just above the RbcL band in order to allow probing of one membrane against two proteins. However, we could enlarge

the visible area to (almost) fully show the RbcL bands in Fig. 8c. We note that RbcL is fully visible in the blot above (which unfortunately had not been stained by coomassie/Revert 700) and RbcL has also been quantified by proteomics (Fig. 8d). All uncropped membranes are displayed in the Source Data file.

4. Also, regarding comments above - how do the relatively mild effects seen in the RNA and protein data correlate with the strong defects in plant physiology?

***Response:** In our view, the effect strengths are in-line with each other. The changes in spliced vs. unspliced RNA pools are quite large, as are the changes in chloroplast-localized proteins of the leaf. Chlorosis could be a direct result, or made worse by photoinhibition, resulting from an imbalance between light energy capture and its utilisation.*

In leaves that had already matured before the induction of silencing, the level of MKIP1 also decreases, but the stability of plastome-encoded proteins in the plastids is likely sufficient to maintain photosynthesis and a green phenotype for the ~1-week period during which MKIP1 levels are reduced. Newly developing leaves, where the photosynthetic machinery must first be established, are more affected, albeit with the expected delay. When we exposed amiR-MKIP1 plants to β -estradiol from the seed or seedling stage, their phenotypes were much more severe, with albino leaves developing from the beginning. However, we did not further analyse such tissues because of the likely pleiotropic effects (see response to comment no. 1, above).

5. The background on group II introns is comprehensive but assumes prior knowledge. Adding a short/brief explanation of their function and relevance in plants could help readers that are unfamiliar with the topic.

***Response:** We expanded the description of the function and relevance of plastidial introns at the beginning of the introduction.*

6. The transition to discussing MatK and its divergence from bacterial maturases is a bit abrupt. Consider smoother transitions.

***Response:** We have added a link to the previous paragraph to make the transition smoother.*

7. MKIP1 gene locus, and gene expression. Notably, the AT3G20440 locus at TAIR database has alternative splicing variants, one of which affects the N-terminal region (AT3G20440.3). Is there a reason that the authors focus on one of these for the genetic analysis AT3G20440.2?

***Response:** In addition to sf2 (At3g20440.2), which we used in our work, there are two other splice form (sf) variants (sf1 [At3g20440.1] and sf3 [At3g20440.3]) annotated on TAIR. The corresponding proteins differ in their middle-to-C-terminal regions (note that At3g20440 is on the minus strand and is therefore read from right to left in genome browsers): Compared to the protein from sf2, the protein from sf3 has a total of seven amino acids inserted and nine deleted between amino acids 365 and 881. The protein from sf1 has a deletion of 30 amino acids after lysine 423. While the mapping coverage of RNA sequencing data fully supports the expression of sf2, it does not support expression of sf1 or sf3 mRNA in diverse tissues, including leaves and roots (see the 'RNA-seq based evidence' section of this locus at <https://jbrowse.arabidopsis.org>). In addition, no peptides specific to proteins from sf1 or sf3*

have been reported in the proteomics database
(<https://www.proteomicsdb.org/proteomicsdb/#protein/proteinDetails/248572/peptides>).

We therefore conclude that sf2 is the predominant, if not the sole, expressed version of MKIP1, which is consistent with its ability to complement the MKIP1 mutant phenotype. We have added an explanation as to why sf2 was selected to the Methods section 'Cloning of plant constructs'.

8. How the authors explain the association of MKIP1 with itself and with the other splicing variants (IP MS data in suppl. data)? do they all correspond to the same peptides? or rather co-IPed the two paralogous variants.

Response: In the IP-MS experiments conducted in Arabidopsis, all MKIP1 peptides detected match to its sf2, i.e. the form that was expressed with a YFP tag and used as bait. Most MKIP1 peptides additionally match to sf1 and/or 3; as described above, they are highly similar to sf2. However, we did not detect any peptides specific to sf1 or sf3. The data thus do not provide evidence for self-association of MKIP1. In relation to a comment of reviewer #1, we also tested self-association of Arabidopsis MKIP1-YFP with MKIP1-FLAG in IP experiments after transient expression in tobacco and could not observe homo-oligomerization (**new Supplementary Fig. 10**).

9. Fig. 1 - Localization analysis. The confocal microscopy requires additional controls (mito's, nuclear at cetera). Note that MKIP1 was identified in a nuclear MS/MS analysis (although may correspond to chloroplast contamination).

Response: We replaced the previous micrograph with more detailed confocal micrographs of Arabidopsis epidermal leaf cells. Using staining by MitoTracker and DAPI, we now also visualise mitochondria and nuclei, in addition to AtMKIP1-YFP and chlorophyll autofluorescence (**Fig. 1b** and **new Supplementary Fig. 3**). Our microscopy analyses indicate that AtMKIP1 exclusively localises to the plastids.

10. Following their data, and in light of Fig 6c (MS results lines 292-294), the authors need to show the global effects of MKIP1 mutations on ribosome biogenesis effects (e.g., sucrose gradients, RNase protection assays of plastidial rRNAs) as well as on chloroplast translation (i.e., MS/MS).

Response: In our initial visualization of rRNAs by gel staining we had only observed unprocessed 23S rRNA resulting from incomplete hidden breaks within the large ribosomal subunit (now in **Fig. 8e**) in AtMKIP1 silenced tissue, and such specific defects on single rRNA species had indeed been linked before to potential primary defects in assembly of the large ribosomal subunit (Reiter et al., 2020). To examine this further, we have now analysed all plastidial rRNAs in detail by northern blotting (**new Supplementary Fig. 14**). These blots show that 1) both AtMKIP1 and AtuL4c silencing leads to aberrant processing of plastidial rRNAs, and 2) that rRNAs of both the small and large plastidial ribosomal subunits are affected. Importantly, this points towards a global plastidial rRNA processing defect, which is a common secondary outcome of mutations affecting plastidial translation: very similar rRNA processing patterns as upon AtMKIP1 silencing have been observed in Arabidopsis mutants of ribosomal proteins of the plastidial large or small ribosomal subunits (Tiller et al., 2012) and the maize ppr5 mutant (Beick et al., 2008), which has a primary defect in tRNA processing and thus secondarily in plastidial translation.

As described in the answers to comment 2, our total proteome analyses points towards a defect in plastidial translation upon AtMKIP1 silencing. This in turn will affect ribosome assembly and, as multiple rRNA processing steps occur after rRNA incorporation into the ribosome, also rRNA processing. However, we stress that these defects are most likely secondary consequences of the group IIA intron splicing defects, and the associated reduction in the abundance of mature tRNAs and ribosomal proteins. The view that the alterations in rRNA processing and plastid translation upon MKIP1 silencing are indirect is consistent with our observation that NtMKIP1 does not co-migrate with a component of the large ribosomal subunit (NtuL4c) in sucrose gradients / size exclusion experiments (new Fig. 4 and Supplementary Fig. 9).

11. The model in Fig 8 is not supported, and might be wrong. Other than a few canonical MAT' s, the association of MatK with group IIA intron RNA targets is indeed currently unknown. However, MatK was previously shown to demonstrate some specificity in its association with its defined gII' s targets in vitro, this in the absent of additional protein cofactors (see Liere, K. and Link, G. (1995) RNA-binding activity of the Matk protein encoded by the chloroplast trnK intron from mustard (Sinapis-Alba L). Nucleic Acids Res, 23, 917-921.; Barthelet, M.M., Pierpont, C.L. and Tavernier, E.K. (2020) Unraveling the role of the enigmatic MatK maturase in chloroplast group IIA intron excision. Plant Direct, 4, e00208. MatK protein contains part of the RT0 sequence motif (a region with importance for specific gII RNA binding), although most of this region (and other T elements) seem indeed missing from MatK.

Response: As discussed in response to Reviewer 1, we recognize that that the model originally presented in Fig. 8, is not yet sufficiently supported and have thus removed it. Further structural studies will be undertaken to determine the mechanism by which MatK associates with its different intron targets and how the other proteins we have identified contribute to its function. We also now mention the earlier evidence that MatK by itself can still bind to RNA to some extent (Barthelet & Hilu, 2008; Liere & Link, 1995) in the discussion. Intron binding is likely conferred by the X domain of MatK, while the contribution of the N-terminal half of this process is unclear. Even though the N-terminus may still have a part of the RT0 sequence and may contribute to RNA binding, conservation of the RT0 is restricted to few amino acids, and these are now predicted with high confidence to make close contacts to MKIP1. Given the general sequence divergence of MatK's N-terminal region and its observed and predicted engagement in protein-protein interactions, we find it plausible that this region does not fulfill its original function anymore, which includes establishing the high-affinity contacts with intron RNA.

Some other minor remarks:

1. The abstract is well-written, but some sentences are lengthy and could benefit from simplification for clarity.

Response: We have adapted the abstract accordingly.

2. The authors may want to briefly explain at the first paragraph in the intro/abstract how At-MKIP1 was originally identified as a MatK interacting partner (co-IPs?)

Response: We have adapted the abstract accordingly.

3. "Arabidopsis MKIP1 specifically co-precipitates all known intron targets of MatK." Could be rephrased (e.g., "In Arabidopsis, MKIP1 co-precipitates with all known intron targets of MatK.")

Response: We have adapted the abstract accordingly.

4. The concluding statement about the N-terminus of MatK could be clearer if rephrased.

Response: We have changed the end of the abstract accordingly.

5. The introduction is quite clear but occasionally verbose. For example: "Chloroplast gene expression involves the splicing of ~20 introns, a process essential for plant viability." Consider to alter and to combine with the subsequent sentence for a smoother transition.

Response: We changed the beginning of the introduction accordingly.

6. The section on MatK's conserved X domain could use reorganization for better logical flow.

Response: We adapted the beginning of the paragraph accordingly.

7. Results line 177 - EMB3120 is annotated at TAIR as - SNOW-WHITE LEAF1, SWL1 (see REF of Yue Wang et al Crucial role of SWL1 in chloroplast biogenesis and development in Arabidopsis thaliana. Plant Cell Rep 43, 135 (2024). <https://doi.org/10.1007/s00299-024-03210-1>.)

Response: We now cite this reference and the first report of the rice homolog (Hayashi-Tsugane et al., 2014) at this instance.

References:

- Beick, S., Schmitz-Linneweber, C., Williams-Carrier, R., Jensen, B., & Barkan, A. (2008). The Pentatricopeptide Repeat Protein PPR5 Stabilizes a Specific tRNA Precursor in Maize Chloroplasts. *Molecular and Cellular Biology*, 28(17), 5337–5347. <https://doi.org/10.1128/mcb.00563-08>
- Hayashi-Tsugane, M., Takahara, H., Ahmed, N., Himi, E., Takagi, K., Iida, S., Tsugane, K., & Maekawa, M. (2014). A mutable albino allele in rice reveals that formation of thylakoid membranes requires the SNOW-WHITE LEAF1 gene. *Plant and Cell Physiology*, 55(1), 3–15. <https://doi.org/10.1093/pcp/pct149>
- Powikrowska, M., Oetke, S., Jensen, P. E., & Krupinska, K. (2014). Dynamic composition, shaping and organization of plastid nucleoids. *Frontiers in Plant Science*, 5(SEP), 1–13. <https://doi.org/10.3389/fpls.2014.00424>

- Reiter, B., Vamvaka, E., Marino, G., Kleine, T., Jahns, P., Bolle, C., Leister, D., & Rühle, T. (2020). The Arabidopsis protein CGL20 is required for plastid 50S ribosome biogenesis. *Plant Physiology*, *182*(3), 1222–1238. <https://doi.org/10.1104/PP.19.01502>
- Tiller, N., Weingartner, M., Thiele, W., Maximova, E., Schöttler, M. A., & Bock, R. (2012). The plastid-specific ribosomal proteins of *Arabidopsis thaliana* can be divided into non-essential proteins and genuine ribosomal proteins. *Plant Journal*, *69*(2), 302–316. <https://doi.org/10.1111/j.1365-313X.2011.04791.x>